# ERROR BOUNDS FOR DEEP LEARNING-BASED UNCERTAINTY PROPAGATION IN SDEs

## ABSTRACT

Stochastic differential equations are commonly used to describe the evolution of stochastic processes. The uncertainty of such processes is best represented by the probability density function (PDF), whose evolution is governed by the Fokker-Planck partial differential equation (FP-PDE). However, it is generally infeasible to solve the FP-PDE in closed form. In this work, we show that physics-informed neural networks (PINNs) can be trained to approximate the solution PDF using existing methods. The main contribution is the analysis of the approximation error: we develop a theory to construct an arbitrary tight error bound with PINNs. In addition, we derive a practical error bound that can be efficiently constructed with existing training methods. Finally, we explain that this error-bound theory generalizes to approximate solutions of other linear PDEs. Several numerical experiments are conducted to demonstrate and validate the proposed methods.

## 1 INTRODUCTION

*Stochastic differential equations* (SDEs) are widely used to model the evolution of stochastic processes across various fields like sciences, engineering, economics, and finance. In many of these applications, particularly in *safety-critical* domains, a key concern is understanding how uncertainty of the process modeled by SDE propagates over space and time. This uncertainty is often represented by a probability density function (PDF) and is governed by the Fokker-Planck partial differential equation (FP-PDE). However, solving the FP-PDE is generally computationally expensive and prone to numerical errors, except in simple cases (Spencer & Bergman, 1993; Drozdov & Morillo, 1996; Tabandeh et al., 2022). Recent advancements suggest using deep-learning frameworks, called *physics-informed neural networks* (PINNs), to approximate PDE solutions with notable success (Sirignano & Spiliopoulos, 2018; Lu et al., 2021). Despite their effectiveness, PINNs are still subject to approximation errors, a crucial concern in safety-critical systems. In this work, we tackle this challenge by developing a method to approximate the PDF of an SDE using PINNs and rigorously bound the approximation error.

Recent works on using PINNs to approximate solutions to PDEs typically analyze approximation errors in terms of total error, representing the cumulative error across all space and time (De Ryck & Mishra, 2022b;a; Mishra & Molinaro, 2023; De Ryck et al., 2024). While this approach may be useful in some applications, it is less informative for SDEs and uncertainty propagation in stochastic processes. Moreover, total error bounds are often overly loose, sometimes exceeding the actual errors by several orders of magnitude. Crucially, these bounds do not provide insight into the worst-case approximation error at specific time instances or within particular subsets of space, which is essential in many stochastic systems. For example, in autonomous driving scenarios involving pedestrian crossings, accurately prediction and bounding the probability of collision requires precise reasoning over specific time instances and spatial regions. Loose over-approximations can lead to undesirable behaviors, such as sudden braking.

In this work, we show how PINNs can be used to approximate PDFs of processes modeled by SDEs and, more importantly, introduce a method for tightly bounding the approximation error as a function of time and space. Our key insight is that the error is related to the residual of the FP-PDE and is governed by the same equation. Thus, a second PINN can be used to learn the error, with its own error also following the FP-PDE. This leads to a recursive formulation of error functions, each of which can be approximated using a PINN. We establish sufficient training conditions under

which this series converges with a finite number of terms. Specifically, we prove that two PINNs are enough to obtain arbitrarily tight error bounds. Additionally, we derive a more practical bound requiring only one error PINN at the cost of losing arbitrary tightness, and provide a method to verify its sufficient condition. Finally, we illustrate and validate these error bounds through experiments on several SDEs, supporting our theoretical claims.

In short, the main contribution is five-fold:

- a method for approximating the PDF of processes modeled by SDEs using PINNs,
- a novel approach to tightly bound the approximation error over time and space through a recursive series of error functions learned by PINNs,
- a proof that this recursive process converges with only two PINNs needed for arbitrarily tight bounds,
- the derivation of a more practical error bound requiring just one PINN, along with a method to verify its sufficiency, and
- validation of the proposed error bounds through experiments on several SDEs.

## 1.1 RELATED WORK

Research on approximating solutions to PDEs using PINNs often focuses on estimating the total error, which represents the cumulative error across all time and space. For instance, (Mishra & Molinaro, 2023) provide an abstract upper bound on the total error, expressed in terms of training error, the number of training samples, and constants related to the stability of PDEs. Their numerical experiments reveal that this total error bound is loose, exceeding the actual errors by nearly three orders of magnitude. Similarly, De Ryck & Mishra (2022a) consider FP-PDE equations deriving from linear stochastic differential equations. They propose an abstract approach to bound the total error in terms of training error and some constants related to the PDEs, but they do not present numerical experiments. In another approach, (De Ryck & Mishra, 2022b) propose a general framework to derive different types of total error bounds for PINNs and operators, while (De Ryck et al., 2024) estimate the total error for Navier-Stokes PDEs. In contrast to these works, this work emphasizes bounding the worst-case error at any specific time. This focus is particularly valuable in practical applications of stochastic systems.

Error analysis is a well-established area focused on demonstrating the approximation capabilities of neural networks. For example, Hornik (1991) proves that a standard multi-layer feed-forward neural network can approximate a target function such that the generalization is arbitrarily small. Yarotsky (2017) considers the worst-case error and shows that deep ReLU neural networks are able to approximate universal functions in the Sobolev space. More recently, deep operator nets (DeepONet) have been suggested to learn PDE operators, with (Lanthaler et al., 2022) proving that for every $\epsilon > 0$, there exists DeepONets such that the total error is smaller than $\epsilon$. While these studies establish that the approximation error (whether in terms of average or worst-case) can be made arbitrarily small, they do not address the critical question: what are the error bounds for a given approximate solution? This is the central issue tackled by this work.

Error estimates have also been studied when neural networks are trained as surrogate models for given target functions. For instance, Barron (1994) derives the total error between given the training configurations and the target function. More recently, Yang et al. (2022) propose to estimate the worst-case approximation error given the target function. A fundamental difference between our work and these studies is that we do not have the target function or model.

Solving PDEs is a well-studied area with various established approaches. For the FP-PDE equation, numerical methods, such as the finite elements method, have been employed (Spencer & Bergman, 1993). Additionally, Chakravorty (2006) uses Galerkin projection method for solution approximation. Recent works (Khoo et al., 2019; Song et al., 2023; Lin & Ren, 2024) present numerical methods for approximating transition probability between two regions, which is also governed by the FP-PDE. For general PDEs, Zada et al. (2021) propose an analytical method to obtain approximate solutions based on optimal auxiliary function. While these studies demonstrate accurate approximations through posterior evaluation, they can be computationally expensive and often lack the ability to quantify and bound the error. In contrast, our method for approximating solutions to the FP-PDE using PINNs is computationally tractable and centers on constructing error bounds for them.

## 2 PROBLEM FORMULATION

The aim of this work is uncertainty propagation with quantified error bounds for continuous time and space stochastic processes using deep neural networks. We specifically focus on stochastic processes described by the following (possibly nonlinear) Stochastic Differential Equations (SDE),

$$d\boldsymbol{x}(t) = f(\boldsymbol{x}(t), t)dt + g(\boldsymbol{x}(t), t)d\boldsymbol{w}(t), \tag{1}$$

where $t \in T \subseteq \mathbb{R}_{\geq 0}$ is time, $\boldsymbol{x}(t) \in X \subseteq \mathbb{R}^n$ is the state of the system at time $t$, and $\boldsymbol{w}(t) \in \mathbb{R}^m$ is a standard Brownian motion. For $\Omega = X \times T$, function $f : \Omega \to \mathbb{R}^n$ represents the deterministic evolution of the system, and function $g : \Omega \to \mathbb{R}^{n \times m}$ is a term that defines the coupling of the noise. We assume that $f(x, t)$ and $g(x, t)$ are locally Lipschitz continuous in $x$, and denote the $i$-th dimension of $f$ and $(j, k)$-th element of $g$ by $f_i$ and $g_{jk}$, respectively. The initial state $\boldsymbol{x}(0)$ is a random variable distributed according to a given probability density function (PDF) $p_0 : X \to \mathbb{R}_{\geq 0}$, i.e., $\boldsymbol{x}(0) \sim p_0$. We assume that $p_0$ is bounded and sufficiently smooth[1].

The solution to the SDE in equation 1 is a stochastic process $\boldsymbol{x}$ with a corresponding PDF $p : \Omega \to \mathbb{R}_{\geq 0}$ over space and time, i.e., $\boldsymbol{x}(t) \sim p(\cdot, t)$ (Øksendal, 2003). PDF $p$ is governed by the Fokker-Planck (FP) partial differential equation (PDE):

$$\frac{\partial p(x, t)}{\partial t} + \sum_{i=1}^{n} \frac{\partial}{\partial x_i}[f_i p(x, t)] - \frac{1}{2} \sum_{i=1, j=1}^{n} \frac{\partial^2}{\partial x_i \partial x_j}\left[\sum_{k=1}^{m} g_{ik} g_{jk} p(x, t)\right] = 0, \tag{2}$$

and must satisfy the initial condition

$$p(x, 0) = p_0(x) \qquad \forall x \in X. \tag{3}$$

To simplify notation, we denote by $\mathcal{D}[\cdot]$ the differential operator associated with the FP-PDE:

$$\mathcal{D}[\cdot] := \frac{\partial}{\partial t}[\cdot] + \sum_{i=1}^{n} \frac{\partial}{\partial x_i}[f_i \cdot] - \frac{1}{2} \sum_{i=1, j=1}^{n} \frac{\partial^2}{\partial x_i \partial x_j}\left[\sum_{k=1}^{m} g_{ik} g_{jk} \cdot\right].$$

Then, equation 2 and equation 3 can be rewritten in a compact form as

$$\mathcal{D}[p(x, t)] = 0, \quad \text{subject to} \quad p(x, 0) = p_0(x). \tag{4}$$

Note that, since $f$ and $g$ are assumed to be locally Lipschitz continuous, the PDE in equation 4 is well-posed, i.e., there exists a sufficiently smooth and unique solution $p$ (Evans, 2022), (Karatzas & Shreve, 2014, Ch. 5, Theorem 2.5).

Computation of $p$ in closed form is generally not possible, and even numerical approaches are limited to simple SDEs (Spencer & Bergman, 1993; Drozdov & Morillo, 1996; Tabandeh et al., 2022). In this work, we focus on using PINNs to approximate $p$, and crucially, we aim to formally bound the resulting approximation error.

**Problem 1** *Given stochastic process $\boldsymbol{x}(t)$ described by the SDE in equation 1, a bounded subset $X' \subset X$, and a time interval $T$, train a neural network $\hat{p}(x, t)$ that approximates $p(x, t)$, and for every $t \in T$ construct $e_B : T \to \mathbb{R}_{\geq 0}$ such that*

$$\sup_{x \in X'} |p(x, t) - \hat{p}(x, t)| \leq e_B(t). \tag{5}$$

In our approach, we exploit the governing equation of $p$ in equation 4 for both training for $\hat{p}$ and for its error quantification. Specifically, we first show that existing methods for training PINNs to approximate solutions of PDEs can be adapted to approximate $p$ well if the training loss is sufficiently small. Then, we show that the resulting approximation error can be written as an infinite series of approximate error functions, each of which satisfying a PDE similar to equation 4. This implies that each error function itself can be approximated using a PINN. Then, we derive conditions, under which only a finite number of such PINNs is needed to obtain an error bound $e_B(t)$ with guarantees.

**Remark 1** *While we focus on $\hat{p}$ being a neural network, our method of deriving temporal error bound $e_B(t)$ is not limited to neural networks and generalizes to any sufficiently smooth function $\hat{p}$ that approximates the true solution $p$.*

---

[1] at least twice continuously differentiable with respect to $x$.

## 3 APPROXIMATING PDF VIA PINN

Given the PDE in equation 4, as common in physics-informed deep learning, we approximate $p$ by learning a neural network $\hat{p}(x, t; \theta)$, where $\theta$ represents the parameters of the neural network. For training, spatial-temporal data points $\{(x_j, 0)_j\}_{j=1}^{N_0}, \{(x_j, t_j)_j\}_{j=1}^{N_r} \subset \Omega$, for some $N_0, N_r \in \mathbb{N}$, are sampled, and the loss function is derived from the governing physics in equation 4 as $\mathcal{L} = w_0 \mathcal{L}_0 + w_r \mathcal{L}_r$, where $w_0, w_r \in \mathbb{R}^+$ are the weights, and

$$\mathcal{L}_0 = \frac{1}{N_0} \sum_{j=1}^{N_0} \|p_0(x_j) - \hat{p}(x_i, 0; \theta)\|_2^2, \qquad \mathcal{L}_r = \frac{1}{N_r} \sum_{j=1}^{N_r} \|\mathcal{D}[\hat{p}(x_j, t_j; \theta)]\|_2^2. \tag{6}$$

The loss function in equation 6 quantifies the deviation of the true and approximate solution in terms of the boundary condition ($\mathcal{L}_0$) and the infinitesimal variation over space and time ($\mathcal{L}_r$) (Sirignano & Spiliopoulos, 2018). The parameters of $\hat{p}(x, t; \theta)$ are learned by minimizing $\theta^* = \arg \min \mathcal{L}$.

**Assumption 1** $\hat{p}$ *is assumed to be at least twice continuously differentiable with respect to $x$ and continuously differentiable with respect to $t$ with bounded derivatives.*

Assumption 1 is present because $\hat{p}$ is trained by the physics-informed loss in equation 6, in which the second term $\mathcal{L}_r$ requires the computation of the first and second derivatives with respect to time and space, respectively. To satisfy Assumption 1, smooth activation functions (e.g., $\mathrm{Tanh}$ and $\mathrm{Softplus}$) can to be used in the architecture of $\hat{p}(x, t; \theta)$. For instance, this assumption is satisfied by a fully connected NN with twice differentiable activation functions.

Our training approach for $\hat{p}$ follows existing methods to approximate PDE solutions using PINNs; see Appendix B for more details. The key difference is that we provide error bounds on the approximation error as detailed in the next section.

## 4 BOUNDING APPROXIMATION ERROR

In this section, we derive bounds for the approximation error $e(x, t) := p(x, t) - \hat{p}(x, t)$. We first characterize $e(x, t)$ as a series of approximate solutions to PDEs. Then, we show that, by training just two PINNs under certain sufficient conditions, the series can be bounded, resulting in arbitrary tight bound on $e(x, t)$. While these conditions are feasible, they may be challenging to verify in practice. To that end, we finally introduce a more practical bound that requires training of only one PINN, albeit at the cost of losing arbitrary tightness. All the proofs are provided in the appendix.

Note that FP-PDE operator $\mathcal{D}$ is a linear operator; hence, by applying it to $e(x, t)$, we obtain:

$$\mathcal{D}[e] = \mathcal{D}[p - \hat{p}] = \mathcal{D}[p] - \mathcal{D}[\hat{p}].$$

As $\mathcal{D}[p] = 0$, we can see that the error is essentially related to the residue of $\mathcal{D}[\hat{p}]$. Then, we can define the governing PDE of $e(x, t)$ as

$$\mathcal{D}[e(x, t)] + \mathcal{D}[\hat{p}(x, t)] = 0 \quad \text{subject to} \quad e(x, 0) = p_0(x) - \hat{p}(x, 0). \tag{7}$$

Hence, using a similar approach as in Section 3, a PINN can approximate $e(x, t)$ in equation 7. Based on this, we can define the $i$-th error and its associated approximation in a recursive manner.

**Definition 1** ($i$-**th error and approximation**) *Let $e_0 := p$ and $\hat{e}_0 := \hat{p}$. We define, for $i \geq 1$, the $i$-th error to be $e_i(x, t) = e_{i-1}(x, t) - \hat{e}_{i-1}(x, t)$, where each $\hat{e}_i$ is a smooth, bounded function that is constructed via a PINN that approximates $e_i$ governed by the recursive PDE (see Appendix A.1):*

$$\mathcal{D}[e_i(x, t)] + \sum_{j=1}^{i} \mathcal{D}[\hat{e}_{j-1}(x, t)] = 0 \quad \text{subject to} \quad e_i(x, 0) = e_{i-1}(x, 0) - \hat{e}_{i-1}(x, 0). \tag{8}$$

By this construction, the approximation error $e(x, t)$, for every choice of $n \geq 0$, is given by

$$e(x, t) = p(x, t) - \hat{p}(x, t) = \sum_{i=1}^{n} \hat{e}_i(x, t) + e_{n+1}(x, t). \tag{9}$$

In the remainder of this section, we derive upper bounds for the right-hand side of equation 9.

First, we express how well $\hat{e}_i$ approximates the $i$-th error $e_i$ by defining the *relative approximation factor* $\alpha_i(t)$ as

$$\alpha_i(t) := \frac{\max_{x \in X'} |e_i(x,t) - \hat{e}_i(x,t)|}{\max_{x \in X'} |\hat{e}_i(x,t)|}. \tag{10}$$

Recall from Def. 1 that $e_i - \hat{e}_i = e_{i+1}$. Hence, equation 10 can be written in a recursive form as

$$\max_{x \in X'} |e_{i+1}(x,t)| = \alpha_i(t) \max_{x \in X'} |\hat{e}_i(x,t)|, \tag{11}$$

which relates the unknown $(i+1)$-th error to the $i$-th error approximation.

**Remark 2** *By the definition of $\alpha_i(t)$ in equation 10, it holds that $\alpha_i(t) \geq 0$ for all $i \geq 1$ and $t \in T$.*

Now let $e_i^*(t), \hat{e}_i^*(t)$ denote the maximum of $e_i(x,t), \hat{e}_i(x,t)$ over subset $X' \subset X$, respectively, i.e.,

$$e_i^*(t) := \max_{x \in X'} |e_i(x,t)|, \quad \hat{e}_i^*(t) := \max_{x \in X'} |\hat{e}_i(x,t)|. \tag{12}$$

Recall that each $\hat{e}_i(x,t)$ can be represented using a PINN. Hence, it is safe to assume that the absolute value of its upper-bound over set $X'$ is strictly greater than zero in finite-time training.

**Assumption 2** *Assume that, for all $1 \leq i < n$, $\hat{e}_i^*(t) > 0$.*

Then, the following lemma upper-bounds the approximation error $e(x,t)$ using $\hat{e}_i^*(t)$.

**Lemma 1** *Consider the approximation error $e(x,t) = p(x,t) - \hat{p}(x,t)$ in equation 9 with $n \geq 2$, and the upper-bounds $\hat{e}_i^*(t)$ for $1 \leq i < n$ over set $X' \subset X$ in equation 12. Define ratio*

$$\gamma_{\frac{i+1}{i}}(t) := \frac{\hat{e}_{i+1}^*(t)}{\hat{e}_i^*(t)}. \tag{13}$$

*Then, under Assumption 2, it holds that, $\forall x \in X'$,*

$$|e(x,t)| \leq \hat{e}_1^*(t)\Big(1 + \sum_{m=2}^{n} \prod_{i=1}^{m-1} \gamma_{\frac{i+1}{i}}(t) + \frac{e_{n+1}^*}{\hat{e}_{n-1}^*} \prod_{i=1}^{n-2} \gamma_{\frac{i+1}{i}}(t)\Big). \tag{14}$$

Next, we derive an upper- and lower-bound for the ratio $\gamma_{\frac{i+1}{i}}(t)$ in equation 14 using $\alpha_i(t)$.

**Lemma 2** *If the relative approximation factors $\alpha_i(t) < 1$ for all $2 \leq i < n$, then*

$$\frac{\alpha_{i-1}(t)}{1 + \alpha_i(t)} \leq \gamma_{\frac{i}{i-1}}(t) \leq \frac{\alpha_{i-1}(t)}{1 - \alpha_i(t)}. \tag{15}$$

Lemma 2 establishes the relationship between ratio $\gamma_{\frac{i}{i-1}}$ and relative approximation factors $\alpha_i$ under condition $\alpha_i < 1$. Intuitively, this condition holds when $\hat{e}_i$ approximates $e_i$ reasonably well (see equation 10). Lastly, we show that under certain conditions on $\alpha_1$ and $\alpha_2$, an ordering over $\gamma_{\frac{2}{1}}, \gamma_{\frac{3}{2}}, \ldots, \gamma_{\frac{i}{i-1}}$ can be achieved.

**Lemma 3** *If, for all $t \in T$,*

$$0 < \alpha_1(t) < 1, \tag{16a}$$

$$0 < \alpha_2(t) < 1 - \alpha_1(t), \tag{16b}$$

$$\alpha_2(t)(1 + \alpha_2(t)) < \alpha_1(t)^2, \tag{16c}$$

*then there exist feasible $0 \leq \alpha_i(t) < 1$ for $2 < i < n$ such that*

$$\gamma_{\frac{i}{i-1}}(t) < \gamma_{\frac{2}{1}}(t) < 1. \tag{17}$$

The intuition behind Lemma 3 is that if $\hat{e}_1$ and $\hat{e}_2$ are trained to certain accuracy (satisfying Conditions 16), then there exist feasible $\hat{e}_3, \hat{e}_4, \ldots, \hat{e}_{n-1}$ such that the ratios $\gamma_{\frac{3}{2}}, \gamma_{\frac{4}{3}}, \ldots, \gamma_{\frac{n-1}{n-2}}$ are upper bounded by $\gamma_{\frac{2}{1}} < 1$. Specifically, Condition 16a on $\alpha_1$ indicates that $\hat{e}_1$ must be learned well enough so that the magnitude of its maximum learning error is less than its own maximum magnitude (see equation 10). By fixing $\alpha_1$, Conditions 16b-16c on $\alpha_2$ require $\hat{e}_2$ to approximate $e_2$ more accurately than the approximation of $e_1$ by $\hat{e}_1$. These conditions are feasible, i.e., they can be satisfied since each PINN can be trained arbitary well (Hornik, 1991; De Ryck et al., 2021; Mertikopoulos et al., 2020; Mishra & Molinaro, 2023). However, verifying them can be challenging. In Section 4.2, we provide a method of checking for $\alpha_1$ condition and derive a bound that only relies on this condition; checking $\alpha_2$ during training remains an open problem.

Finally, we can state our main result, which is a bound on the approximation error of $\hat{p}$ using Lemmas 1-3. Specifically, the following theorem shows that the approximation error bound in Lemma 1 becomes a geometric series as $n \to \infty$ under Conditions 16; hence, solving Problem 1.

**Theorem 1 (Temporal error bound)** *Consider Problem 1 and two approximate error functions $\hat{e}_1(x, t), \hat{e}_2(x, t)$ constructed by Definition 1 that satisfy Conditions 16. Then,*

$$|p(x,t) - \hat{p}(x,t)| \leq e_B(t) = \hat{e}_1^*(t)\Big(\frac{1}{1 - \gamma_{\frac{2}{1}}(t)}\Big), \tag{18}$$

*where $\hat{e}_1^*(t)$ is defined in equation 12, and $\gamma_{\frac{2}{1}}(t) = \hat{e}_2^*(t)/\hat{e}_1^*(t)$.*

The above theorem shows that temporal error bound $e_B(t)$ can be obtained by training only two PINNs that approximate the first two errors $e_1, e_2$ according to Def. 1 and that satisfy Conditions 16. In fact, using these two PINNs, it is possible to construct an arbitrary tight $e_B$ as stated below.

**Theorem 2 (Temporal error bound of arbitrary tightness)** *Given Problem 1 and tolerance $\epsilon \in (0, \infty)$ on the error bound, a temporal error bound $e_B(t)$ can be obtained by training two approximate error functions $\hat{e}_1(x, t)$ and $\hat{e}_2(x, t)$ through physics-informed learning such that*

$$e_B(t) - \max_{x \in X'} |e(x,t)| < \epsilon. \tag{19}$$

The proof of Theorem 2 is based on the observation that $\gamma_{\frac{2}{1}} \to 0$ when (i) $\hat{e}_1(x, t) \to e_1(x, t)$ and (ii) $\hat{e}_2(x, t) \to e_2(x, t)$. Then, according to equation 18, $e_B(t) \to \hat{e}_1^*(t)$, which itself $\hat{e}_1^*(t) \to e_1^*(t)$ under (i). Since, PINNs $\hat{e}_1$ and $\hat{e}_2$ can be made arbitrary well, $e_B$ can be arbitrary tight. This result is important because it shows that arbitrary tightness can be achieved without the need for training infinite number of PINNs, i.e., $\hat{e}_i, i = 1, 2, \ldots$

**Remark 3** *The construction of $e_B(t)$ in Theorems 1 only requires the values of $\hat{e}_1^*(t)$ and $\gamma_{\frac{2}{1}}(t)$ which are obtained from the known functions $\hat{e}_1(x, t), \hat{e}_2(x, t)$. Checking for $\alpha_1$ and $\alpha_2$ conditions can be performed a posterior.*

**Remark 4** *Given the approximate functions $\hat{p}$ and $\hat{e}_1$, temporal bound $e_B$ becomes tighter as the approximation accuracy of $\hat{e}_2$ increases. As $\hat{e}_2 \to e_2$, $\alpha_2 \to 0^+$. Also, as $\alpha_2 \to 0^+$, by equation 15, the upper bound of $\gamma_{\frac{2}{1}}$ decreases, and consequently, $e_B$ becomes tighter by equation 18.*

In the following subsections, we extend the result of Theorem 1 which is based on training $n = 2$ approximate error PINNs, to cases of $n > 2$ and $n = 1$ to bound error of $\hat{p}$.

### 4.1 $n$-TH ORDER SPACE-TIME ERROR BOUND ($n > 2$)

Here, we derive a generalized error bound for $e(x, t)$ with approximation error PINNs $\hat{e}_i$, where $i = 1, \ldots, n$ for $n > 2$. Note that an alternative way to express the error bound in Theorem 1 is as an interval $e(x, t) \in \big[-e_B(t), e_B(t)\big]$, which is uniform over $x$ for any $t \in T$. Below, we show that, for $n > 2$, an error bound that depends on both space and time can be constructed.

**Corollary 1 (Space-time Error Bound)** *Consider PINNs $\hat{e}_i(x,t)$, $i = 1, \ldots, n$, for some $n > 2$ trained per Def.1 such that $\alpha_{n-1}$ and $\alpha_n$ satisfy Conditions 16, and define the $n$-th order temporal error bound to be*

$$e_B^n(t) = \hat{e}_{n-1}^*(t)\left(\frac{1}{1 - \gamma_{\frac{n}{n-1}}(t)}\right),$$

*where $\hat{e}_{n-1}^*(t)$ is defined in equation 12, and $\gamma_{\frac{n}{n-1}}(t) = \hat{e}_n^*(t)/\hat{e}_{n-1}^*(t)$. Then,*

$$e(x,t) \in \left[\sum_{i=1}^{n-2} \hat{e}_i(x,t) - e_B^n(t), \ \sum_{i=1}^{n-2} \hat{e}_i(x,t) + e_B^n(t)\right]. \tag{20}$$

This corollary shows that, even though 2-nd order error approximation is sufficient to obtain a temporal bound (Theorem 1), higher order approximations lead to more information, i.e., space in addition to time, on the error bound.

### 4.2 First Order Temporal Error Bound ($n = 1$)

We also present a temporal error bound by learning only the first error approximation function $\hat{e}_1$, which removes the dependence on $\alpha_2$ at the cost of losing the arbitrary tightness property.

**Corollary 2 (First order temporal error bound)** *Let $\hat{e}_1$ be trained such that $\alpha_1(t) < 1$ for all $t \in T$. Then*

$$|e(x,t)| < e_S(t) = 2\hat{e}_1^*(t). \tag{21}$$

Note that, while the first-order error bound $e_S(t)$ is at most twice larger than the arbitrary tight error bound $e_B(t)$ in Theorem 1, it has significant practical uses. Firstly, it only requires training of one PINN, i.e., $\hat{e}_1$. Secondly, the condition $\alpha_1(t) < 1$ can be checked during training of $\hat{e}_1$ using properties of the FP-PDE as detailed below.

**Checking $\alpha_1(t) < 1$ condition** From the definition of $\alpha_1(t)$ in equation 10, it suffices to bound the unknown term $|e_1(x,t) - \hat{e}_1(x,t)|$ for all $(x,t) \in \Omega$ to check for $\alpha_1$. We do this by using three constants: two related to FP-PDE as introduced in (Mishra & Molinaro, 2023), and one universal constant from Sobolev embedding theorem (Mizuguchi et al., 2017)(Hunter & Nachtergaele, 2001, Theorem 12.71). First, the *stability* constant $C_{pde}$ of the first error PDE ($\mathcal{D}[\cdot] + \mathcal{D}[\hat{p}]$) is defined as

$$\|e_1(x,t) - \hat{e}_1(x,t)\|_Z \leq C_{pde}\|(\mathcal{D}[e_1] + \mathcal{D}[\hat{p}]) - (\mathcal{D}[\hat{e}_1] + \mathcal{D}[\hat{p}])\|_Y,$$

where $Z = W^{k,q}$ norm , $Y = L^s$ norm, $1 \leq s, q < \infty$, and $k \geq 0$. Note that since $e_1, \hat{e}_1$ and $(\mathcal{D}[e_1] + \mathcal{D}[\hat{p}]) - (\mathcal{D}[\hat{e}_1] + \mathcal{D}[\hat{p}]) = -(\mathcal{D}[\hat{e}_1] + \mathcal{D}[\hat{p}])$ are bounded[2], such constant $C_{pde}$ exists for $k \leq 1$. Second, the *quadrature* constant $C_{quad} > 0$ is defined such that for some $\beta > 0$,

$$\left|\int_\Omega \left(\mathcal{D}[\hat{e}_1(x,t)] + \mathcal{D}[\hat{p}(x,t)]\right)dxdt - \sum_{i=1}^N w_i\left(\mathcal{D}[\hat{e}_1(x_i,t_i)] + \mathcal{D}[\hat{p}(x_i,t_i)]\right)\right| \leq C_{quad}N^{-\beta},$$

where $\{(x_i,t_i)_i\}_{i=1}^N \in \Omega$ is a set of $N$ quadrature points, and $w_i \in \mathbb{R}_{>0}$ are weights according to the quadrature rules. The procedure of deriving these universal constants for general PDEs with bounded derivatives is shown in (Mishra & Molinaro, 2023). The third constant $C_{embed}$ is defined as

$$\|e_1(x,t) - \hat{e}_1(x,t)\|_\infty \leq C_{embed}\|e_1(x,t) - \hat{e}_1(x,t)\|_{W^{1,q}}.$$

Constant $C_{embed}$ exists because $e_1(x,t) - \hat{e}_1(x,t)$ is bounded (per Def. 1), and the first derivatives of $e_1(x,t)$ and $\hat{e}_1(x,t)$ are also bounded.

**Proposition 1 (Checking $\alpha_1(t) < 1$)** *Let $x \in \mathbb{R}^n$, $\{(x_i,t_i)_i\}_{i=1}^N \in \Omega$ be $N$ space-time samples based on quadrature rules, $\hat{e}_1(x,t)$ be the first error approximation, and let $\varepsilon_T$ be the physics-informed loss of $\hat{e}_1(x,t)$ evaluated on the set $\{(x_i,t_i)_i\}_{i=1}^N$. Then for some $q \geq 2$ and $\beta > 0$, $\alpha_1(t) < 1$ for all $t \in T$ if*

$$\frac{1}{\min_t \hat{e}_1^*(t)}\left[C_{embed}\left(C_{pde}\varepsilon_T + C_{pde}C_{quad}^{\frac{1}{q}}N^{\frac{-\beta}{q}}\right)\right] < 1. \tag{22}$$

---

[2] $\hat{p}, \hat{e}_1$ are approximate functions with bounded derivatives

By Proposition 1, it is clear that as the training loss decreases ($\varepsilon_T \to 0$) with sufficiently large number of samples ($N \to \infty$), the left-hand side of equation 22 goes to zero. Hence, condition $\alpha_1 < 1$ can be satisfied by training with a sufficiently large dataset and small loss.

**Remark 5 (Generalization to linear PDEs)** *While the presented approach focuses on SDEs and training an approximate PDF $\hat{p}$ and bounding its error, the only essential requirement is that the FP-PDE operator $\mathcal{D}$ is linear. Therefore, this approach naturally extends to all linear PDEs (linear $\mathcal{D}$) subject to initial and boundary conditions. We illustrate this in a case study in Sec. 5.*

# 5 NUMERICAL EXPERIMENTS

We present illustrative experiments to demonstrate the proposed methods on ten systems listed in Table 1. The table indicates the method to obtain the *true* solution. Note that the '1D Heat PDE' system is an illustration of generalizability of our method to linear PDEs beyond SDEs. We also note that these experiments are not an exhaustive study on hyperparameters or neural network architecture but aim to showcase the efficacy of the error bounds using existing PINN training methods. All the details on the system dynamics, hyperparameters, additional plots, etc. are provided in Appendix B.

Table 1: Systems dynamics with their initial conditions (I.C.) and true solution method. Computation time for Monte-carlo simulations are reported. The parameters for high-dimensional systems (3D-10D) are provided in Appendix B.6

| System | Dynamics | I.C. | True Solution |
|---|---|---|---|
| 1D Linear SDE | $dx = -0.2x dt + \sqrt{0.4} dw$ | Gaussian | analytical |
| 1D Nonlinear SDE | $dx = (-0.1x^3 + 0.1x^2 + 0.5x + 0.5)dt + 0.8dw$ | $\mathcal{N}(-2, 0.5^2)$ | Monte-Carlo (100 hrs) |
| 1D State-dependent SDE | $dx = (0.002x)dt + (0.01x)dw$ | Gaussian | analytical |
| Inverted Pendulum SDE | $dx = \begin{bmatrix} x_2 \\ -\sin(x_1) \end{bmatrix} dt + \begin{bmatrix} 0.5 & 0.0 \\ 0.0 & 0.5 \end{bmatrix} dw$ | $\mathcal{N}(\begin{bmatrix} 0.5\pi \\ 0.0 \end{bmatrix}, \begin{bmatrix} 0.5 & 0.0 \\ 0.0 & 0.5 \end{bmatrix})$ | Monte-Carlo (13 hrs) |
| 1D Heat PDE | $u_t - u_{xx} = 0$ | $-\sin(\pi x)$ | analytical |
| 3D OU | $dx = (A_3 x)dt + B_3 dw$ | $\mathcal{N}(\mu_3, \Sigma_3)$ | Numerical integration |
| 3D Time-varying OU | $dx = (\tilde{A}_3(t)x)dt + B_3 dw$ | $\mathcal{N}(\mu_3, \Sigma_3)$ | Numerical integration |
| 7D OU | $dx = (A_7 x)dt + B_7 dw$ | $\mathcal{N}(\mu_7, \Sigma_7)$ | Numerical integration |
| 10D OU | $dx = (A_{10} x)dt + B_{10} dw$ | $\mathcal{N}(\mu_{10}, \Sigma_{10})$ | Numerical integration |
| 10D Time-varying OU | $dx = (\tilde{A}_{10}(t)x)dt + B_{10} dw$ | $\mathcal{N}(\tilde{\mu}_{10}, \Sigma_{10})$ | Numerical integration |

Our implementation is in Python and Pytorch, and the code is provided in the supplementary material. All experiments are conducted on a MacBook Pro with Apple M2 processor and 24GB RAM, excepts for the multiple trials on the '1D nonlinear SDE', which was run on an AMD Ryzen 5 6-Core Processor with 32GB RAM and NVIDIA GeForce RTX 2060.

Table 2: Error bound results. Here, $t^{\hat{p}}_{train}$ and $t^{\hat{e}_1}_{train}$ are the training times in seconds, $e_S^{\max} := \max_t(e_S(t)/\max_x p(x,t))$ and $e_S^{\text{avg}} := \text{avg}_t(e_S(t)/\max_x p(x,t))$ are the maximum and average of the first temporal error bound $e_S$ normalized by the true solution, $\text{Gap}^{\min} := \min_t((e_S(t) - e^*(t))/\max_x p(x,t))$ and $\text{Gap}^{\max} := \max_t((e_S(t) - e^*(t))/\max_x p(x,t))$ are the minimum and maximum gaps (over time) between the error bound and maximum error normalized by the true solution, $\alpha_1^{\max} := \max_t \alpha_1(t)$, and $\alpha_1^{\text{var}} := \text{var}_t \alpha_1(t)$. Each row is the result of one random seed.

| System | $\hat{p}$ loss | $\hat{e}_1$ loss | $t^{\hat{p}}_{\text{train}}$ | $t^{\hat{e}_1}_{\text{train}}$ | $e_S^{\max}$ | $e_S^{\text{avg}}$ | $\text{Gap}^{\min}$ | $\text{Gap}^{\max}$ | $\alpha_1^{\max}$ | $\alpha_1^{\text{var}}$ |
|---|---|---|---|---|---|---|---|---|---|---|
| 1D Linear SDE | 2e-3 | 2e-2 | 5 | 17 | 0.19 | 0.18 | 0.064 | 0.085 | 0.37 | 1e-3 |
| 1D Nonlinear SDE | 1e-3 | 4e-3 | 718 | 3723 | 0.48 | 0.27 | 0.054 | 0.214 | 0.60 | 6e-3 |
| 1D Nonlinear SDE (GPU, seed0) | 1e-4 | 4e-3 | 345 | 4433 | 0.14 | 0.05 | 0.007 | 0.062 | 0.45 | 4e-3 |
| 1D State-dependent SDE | 5e-3 | 5e-3 | 31 | 598 | 0.24 | 0.14 | 0.026 | 0.130 | 0.43 | 6e-3 |
| Inverted Pendulum SDE | 1e-3 | 4e-2 | 1411 | 3576 | 0.25 | 0.16 | 0.015 | 0.132 | 0.75 | 3e-2 |
| 1D Heat PDE | 1e-4 | 4e-5 | 41 | 156 | 135 | 10.3 | 0.002 | 49.10 | 0.40 | 5e-3 |
| 3D OU | 1e-4 | 8e-3 | 276 | 2017 | 0.05 | 0.04 | 0.015 | 0.029 | 0.20 | 2e-4 |
| 3D Time-varying OU | 1e-4 | 4e-3 | 338 | 2219 | 0.06 | 0.05 | 0.020 | 0.032 | 0.16 | 3e-4 |
| 7D OU | 2e-4 | 1e-2 | 1018 | 2684 | 0.19 | 0.11 | 0.036 | 0.098 | 0.74 | 2e-2 |
| 10D OU | 1e-4 | 1e-2 | 1710 | 3670 | 0.20 | 0.15 | 0.067 | 0.119 | 0.68 | 5e-3 |
| 10D Time-varying OU | 1e-4 | 6e-3 | 2835 | 13883 | 0.16 | 0.12 | 0.053 | 0.095 | 0.98 | 9e-3 |

For the 1D Nonlinear SDE on GPU, the variance of $\alpha_1$ over all six random seeds $i = \{0, 1, ..., 5\}$ is $\text{var}_{t,i}\alpha_1^{(i)}(t) = 0.11$.

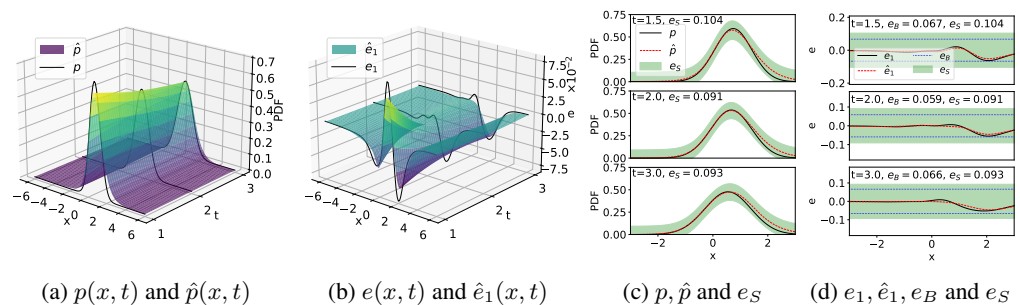

(a) $p(x,t)$ and $\hat{p}(x,t)$     (b) $e(x,t)$ and $\hat{e}_1(x,t)$     (c) $p, \hat{p}$ and $e_S$     (d) $e_1, \hat{e}_1, e_B$ and $e_S$

Figure 1: True and approximate PDF solutions for the 1D Linear SDE with quantified error bounds.

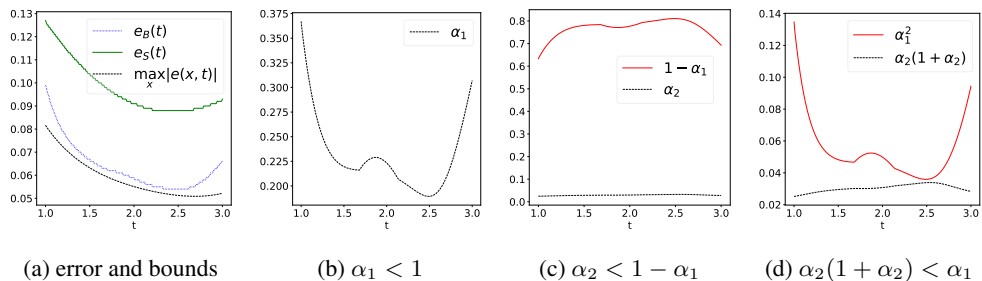

(a) error and bounds     (b) $\alpha_1 < 1$     (c) $\alpha_2 < 1 - \alpha_1$     (d) $\alpha_2(1+\alpha_2) < \alpha_1$

Figure 2: Error $e$ and the first- and second-order temporal bounds $e_S, e_B$, along with the training conditions of $\alpha_1(t)$ and $\alpha_2(t)$ in equation 16 for all $t \in T$ of the 1D Linear SDE.

Table 2 summarizes the results on all systems. Note that the smaller $e_S^{\max}$ and $e_S^{\text{avg}}$ are, the tighter error bounds are. Positive $\text{Gap}^{\min}$ implies that the bound is valid, small $\text{Gap}^{\max}$ implies that the bound is close to the true error, and the smaller $\alpha_1^{\max}$ is, the better $\hat{e}_1$ is trained. We also note that the 1D Heat experiment shows large values of the normalized metrics $e_S^{\max}$, $e_S^{\text{avg}}$, and $\text{Gap}^{\min}$ because its true solution (used in the denominator) becomes extremely small at the final time.

In summary, Table 2 shows: (i) *scalability*: our framework is able to scale to 10-dimensional system, (ii) *stability*: the variance on $\alpha_1$ over the time domain is small, showing the error bound's applicability for all time, (iii) *training challenges*: training of $\hat{e}_1$ may encounter local minima due to random initialization of neural networks, (iv) $\alpha_1$ *condition*: $\alpha_1 < 1$ is satisfied though it becomes increasingly challenging to meet as dimensionality grows, and (v) *error bound tightness*: the error bounds are tight across all dynamical systems. Below, we discuss individual systems in more details.

**1D Linear SDE** Figs. 1a-1b visualize the true and learned PDFs $p$ and $\hat{p}$ and the true and learned errors $e$ and $\hat{e}_1$, respectively. PDFs $p$ and $\hat{p}$ along with error bound $e_S(t)$ at $t = 1.5, 2, 3$ seconds are shown in Fig. 1c. Observe that $p$ is always within $e_S$ bound from $\hat{p}$, validating the bound. Fig. 1d shows errors $e$ and $\hat{e}_1$ and compares bound $e_S(t)$ with the arbitrary tight error bound $e_B(t)$ at the same time instances. As predicted, $e_B(t)$ is tighter than $e_s(t)$. We note that learning $\hat{e}_2$ is challenging; hence, for illustration purposes of $e_B(t)$, we used $\hat{e}_2 = e_2 + \delta$, where $\delta$ is a small perturbation for this experiment. Fig. 2 provides a different visualization for $e_S(t)$ and $e_B(t)$ as well as satisfaction of the $\alpha_1$ and $\alpha_2$ conditions. Specifically, Fig. 2a validates that $\max_x |e(x,t)| \leq e_B(t) \leq e_S(t)$ for all $t \in T$. Note that $e_S(t)/e_B(t)$ is at most $1.63 < 2$, as predicted by Corollary 2.

**1D Nonlinear SDE** Figs. 3a-3b show the PDFs $p$ and $\hat{p}$ and errors $e$ and $\hat{e}_1$. The error bound $e_S(t)$ is illustrated in Figs. 3c-3d in the solution and error spaces, respectively. Observe that the true error is upper bounded, and the true PDF $p$ lies within $e_S$ of approximate PDF $\hat{p}$. Figs. 3e-3f show a tighter $e_S(t)$ by training neural networks (with more complicated activation functions) on GPU. To illustrate that $\alpha_1$ does in fact decrease with more training, we conducted multiple training trials for this system. Fig. 3g shows the obtained results, validating that $\alpha_1$ does indeed decrease as the training loss of the $\hat{e}_1(x,t)$ decreases, as predicted by Proposition 1. Note that one trial (out of six trials) failed to train $\hat{e}_1$ that satisfies $\alpha_1(t) < 1$ for some $t$, as seen in Fig. 9 in Appendix B.2.

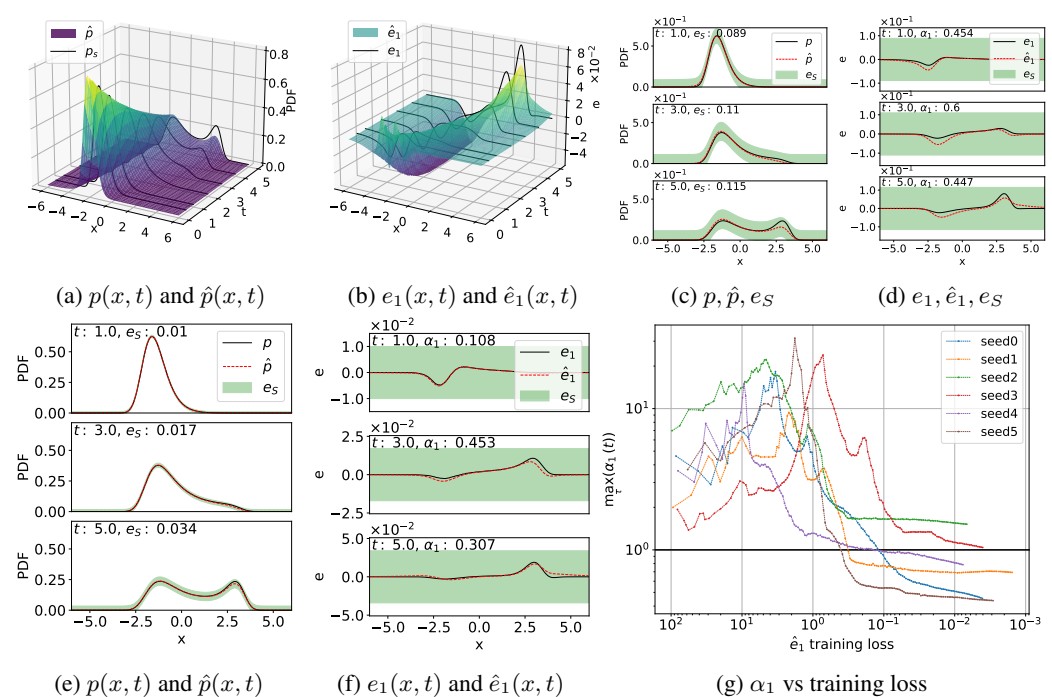

(a) $p(x,t)$ and $\hat{p}(x,t)$     (b) $e_1(x,t)$ and $\hat{e}_1(x,t)$     (c) $p, \hat{p}, e_S$     (d) $e_1, \hat{e}_1, e_S$

(e) $p(x,t)$ and $\hat{p}(x,t)$     (f) $e_1(x,t)$ and $\hat{e}_1(x,t)$     (g) $\alpha_1$ vs training loss

Figure 3: Visualization of the results for 1D Nonlinear SDE. (a)-(d) illustrate error bound $e_S$, and (e)-(f) show one (seed0) of the multiple training trials on GPU, (g) $\alpha_1^{\max}$ is plotted vs $\hat{e}$ loss.

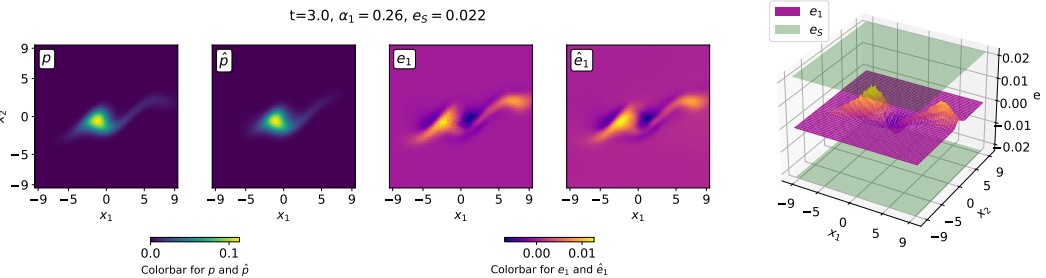

Figure 4: First order temporal error bound of the 2D inverted pendulum at $t = 3$. At the right: the approximation error $e$ is bounded by the green 3D surface $e_S$ constructed by $\hat{e}_1$.

**Others** Fig. 4 visualizes error bound $e_S$ for the 2D inverted pendulum at a given time, showing $e_S$ for multi-dimensional systems. See Appendices B.3-B.6 for results of other systems.

## 6 CONCLUSION

We introduced a physics-informed learning method to approximate the PDF of an SDE and bound its error using a series of recursive error functions learned with PINNs. We proved that only a finite number of recursive steps are required to bound the error, with two error terms being sufficient to achieve arbitrarily tight bounds at any time instance. We also developed a more efficient approach by constructing a first order temporal error bound using just one error function, which reduces computation, provides clear termination criteria, and yields bounds at most twice as loose as the tightest ones. This method was validated on several non-Gaussian dynamical systems. In our implementation, we trained the solution and error functions separately but hypothesize that jointly training them could improve performance and reliability. Future work will explore this joint training approach.

## REPRODUCIBILITY STATEMENT

All the results can be reproduced via the supplemental zip file. There are two folders in the zip file: (1) pinn_pde-release-2025ICLR , and (2) pinn_pde-release-2025ICLR_GPU. The former folder containes the main results that are built on the Macbook Pro. The latter folder includes the results that are built using the Linux desktop. Both folders contain a README.md file that explains the steps of building and running the python codes. Python virtual environments are used to manage the required packages; they are listed in the requirements.txt file. It is recommended that the exact same packages with same versions are installed for reproducibility purpose. The pre-trained neural networks used to generate the results of this paper are provided. One can use these pre-trained neural networks to reproduce the plots by passing the $--$train$= 0$ argument. The code are designed to use the same random seeds, so one can also train the exact same neural networks by passing the $--$train$= 1$ argument, assuming that the required packages are installed successfully.

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

## A   PROOFS

### A.1   DERIVATION OF DEFINITION 1

Denote $e(x,t) := e_1(x,t) = p(x,t) - \hat{p}(x,t)$ as the first error and initialize $e_0(x,t) := p(x,t)$ and $\hat{e}_0(x,t) = \hat{p}(x,t)$. Then, Eq. equation 7 becomes Definition 1 for $i = 1$:

$$\mathcal{D}[e_1(x,t)] + \mathcal{D}[\hat{e}_0(x,t)] = 0, \quad \text{subject to } e_1(x,0) = e_0(x,0) - \hat{e}_0(x,0).$$

For $i = 2$, we define $e_2(x,t) := e_1(x,t) - \hat{e}_1(x,t)$ and obtain $\mathcal{D}[e_2(x,t)] = \mathcal{D}[e_1(x,t)] - \mathcal{D}[\hat{e}_1(x,t)]$ (because $\mathcal{D}[\cdot]$ is a linear operator). Since $\hat{e}_1 \neq e_1$, we have

$$\mathcal{D}[\hat{e}_1] + \mathcal{D}[\hat{e}_0] := r_1 \neq 0.$$

Hence, we have the recursive PDE for $i = 2$ (omitting $x$ and $t$ for simplicity of presentation):

$$\mathcal{D}[e_2] = \mathcal{D}[e_1] - \mathcal{D}[\hat{e}_1] = (-\mathcal{D}[\hat{e}_0]) - (-\mathcal{D}[\hat{e}_0] + r_1) = -r_1 \implies \mathcal{D}[e_2] + r_1 := \mathcal{D}[e_2] + \sum_{j=1}^{2} \mathcal{D}[\hat{e}_{j-1}] = 0.$$

The derivation recursively follows for $i > 2$.

### A.2   PROOF OF LEMMA 1

**Proof 1** *From Definition 1, we have that, for all $x \in X'$,*

$$|p(x,t) - \hat{p}(x,t)| = \left| \sum_{i=1}^{n} \hat{e}_i(x,t) + e_{n+1}(x,t) \right| \leq \sum_{i=1}^{n} |\hat{e}_i(x,t)| + |e_{n+1}(x,t)|$$

$$\leq \sum_{i=1}^{n} \max_x |\hat{e}_i(x,t)| + \max_x |e_{n+1}(x,t)| := \sum_{i=1}^{n} \hat{e}_i^*(x,t) + e_{n+1}^*(x,t).$$

*From the definition of $\gamma_{\frac{i+1}{i}}$ in equation 13, we obtain (omitting $t$ for simplicity of presentation)*

$$|p(x,\cdot) - \hat{p}(x,\cdot)| \leq \hat{e}_1^*\left(1 + \frac{\hat{e}_2^*}{\hat{e}_1^*} + \frac{\hat{e}_3^*}{\hat{e}_1^*} + \cdots + \frac{\hat{e}_n^*}{\hat{e}_1^*} + \frac{e_{n+1}^*}{\hat{e}_1^*}\right)$$

$$= \hat{e}_1^*\left[1 + \gamma_{\frac{2}{1}} + \gamma_{\frac{2}{1}}\gamma_{\frac{3}{2}} + \cdots + (\gamma_{\frac{2}{1}}\gamma_{\frac{3}{2}}\ldots\gamma_{\frac{n}{n-1}}) + (\gamma_{\frac{2}{1}}\gamma_{\frac{3}{2}}\ldots\gamma_{\frac{n-1}{n-2}}\gamma_{\frac{n}{n-1}}\frac{e_{n+1}^*}{\hat{e}_n^*})\right]$$

$$:= \hat{e}_1^*\left[1 + \gamma_{\frac{2}{1}} + \gamma_{\frac{2}{1}}\gamma_{\frac{3}{2}} + \cdots + (\gamma_{\frac{2}{1}}\gamma_{\frac{3}{2}}\ldots\gamma_{\frac{n}{n-1}}) + (\gamma_{\frac{2}{1}}\gamma_{\frac{3}{2}}\ldots\gamma_{\frac{n-1}{n-2}}\frac{\hat{e}_n^*}{\hat{e}_{n-1}^*}\frac{e_{n+1}^*}{\hat{e}_n^*})\right]$$

$$= \hat{e}_1^*\left[1 + \gamma_{\frac{2}{1}} + \gamma_{\frac{2}{1}}\gamma_{\frac{3}{2}} + \cdots + (\gamma_{\frac{2}{1}}\gamma_{\frac{3}{2}}\ldots\gamma_{\frac{n}{n-1}}) + (\gamma_{\frac{2}{1}}\gamma_{\frac{3}{2}}\ldots\gamma_{\frac{n-1}{n-2}}\frac{e_{n+1}^*}{\hat{e}_{n-1}^*})\right].$$

### A.3 PROOF OF LEMMA 2

**Proof 2** *From Definition 1, we have, for $i \geq 0$,*
$$e_i(x,t) = \hat{e}_i(x,t) + e_{i+1}(x,t). \tag{23}$$
*By taking the maximum on the absolute value of equation 23, we get*
$$\max_x |e_i(x,t)| \leq \max_x |\hat{e}_i(x,t)| + \max_x |e_{i+1}(x,t)|. \tag{24}$$
*Similarly, from equation 23, we obtain*
$$\hat{e}_i(x,t) = e_i(x,t) - e_{i+1}(x,t) \implies$$
$$\max_x |\hat{e}_i(x,t)| \leq \max_x |e_i(x,t)| + \max_x |e_{i+1}(x,t)|. \tag{25}$$
*Now take $2 \leq i < n$, and suppose the corresponding $\alpha_i(t) < 1$. Then, we can write the two inequalities in Eqs. equation 24 and equation 25 with the definition of $\hat{e}_i^*(t)$ in equation 12 and the expression in equation 11 as*
$$\begin{cases} \alpha_{i-1}(t)\hat{e}_{i-1}^*(t) \leq \hat{e}_i^*(t) + \alpha_i(t)\hat{e}_i^*(t) \\ \hat{e}_i^*(t) \leq \alpha_{i-1}(t)\hat{e}_{i-1}^*(t) + \alpha_i(t)\hat{e}_i^*(t). \end{cases} \tag{26}$$
*By rearranging equation 26, we obtain the lower and upper bounds of $\gamma_{\frac{i}{i-1}}(t)$:*
$$\frac{\alpha_{i-1}(t)}{1 + \alpha_i(t)} \leq \frac{\hat{e}_i^*(t)}{\hat{e}_{i-1}^*(t)} = \gamma_{\frac{i}{i-1}}(t) \leq \frac{\alpha_{i-1}(t)}{1 - \alpha_i(t)}, \; 2 \leq i < n, \tag{27}$$
*which is well defined because the denominator $\hat{e}_{i-1}^* = e_{n-2}^* > 0$ by Assumption 2, and the (RHS) of equation 27 is always $\geq$ the (LHS) of equation 27 if $0 \leq \alpha_i(t) < 1$ for all $2 \leq i < n$.*

### A.4 PROOF OF LEMMA 3

**Proof 3** *For simplicity of presentation, we omit writing the dependent variable $t$. Assume the conditions in equation 16 are satisfied; then it is true that $0 < \alpha_2 < 1$. Since both $\alpha_1, \alpha_2 < 1$, by Lemma 2 and Condition equation 16b, we obtain*
$$\gamma_{\frac{2}{1}} \leq \frac{\alpha_1}{1 - \alpha_2} < 1,$$
*proving the RHS of equation 17.*

*For the LHS of equation 17, let $\alpha_i \leq \alpha_2$ for all $2 < i < n$. Since $\alpha_2 < 1$, then by Lemma 2, we have*
$$\gamma_{\frac{i}{i-1}} \leq \frac{\alpha_{i-1}}{1 - \alpha_i} \leq \frac{\alpha_{i-1}}{1 - \alpha_2} \leq \frac{\alpha_2}{1 - \alpha_2}. \tag{28}$$
*What remains is to show that RHS of equation 28 is $< \gamma_{\frac{2}{1}}$. From Condition equation 16c, we have*
$$\alpha_2(1 + \alpha_2) < \alpha_1^2 \tag{29}$$
$$< \alpha_1(1 - \alpha_2), \tag{30}$$
*where equation 30 holds by Condition equation 16b. From equation 30, we obtain*
$$\frac{\alpha_2}{1 - \alpha_2} < \frac{\alpha_1}{1 + \alpha_2}. \tag{31}$$
*By combining Eqs. equation 28 and equation 31, we have*
$$\gamma_{\frac{i}{i-1}} < \frac{\alpha_1}{1 + \alpha_2} < \gamma_{\frac{2}{1}}, \; 2 < i < n.$$

## A.5 Proof of Theorem 1

**Proof 4** *Take $n \to \infty$ for Lemma 1, and train $\hat{e}_1$ and $\hat{e}_2$ such that the sufficient conditions of equation 16 are met, therefore, $\gamma_{\frac{3}{2}}, \gamma_{\frac{4}{3}}, \ldots, \gamma_{\frac{n-1}{n-2}} < \gamma_{\frac{2}{1}} < 1$ by Lemma 3. Then we have*

$$
|p(x,t) - \hat{p}(x,t)|
$$

$$
\leq \hat{e}_1^* \lim_{n\to\infty} \left( 1 + \gamma_{\frac{2}{1}} + \gamma_{\frac{2}{1}}\gamma_{\frac{3}{2}} + \cdots + \left[ \gamma_{\frac{2}{1}}\gamma_{\frac{3}{2}} \ldots \gamma_{\frac{n-1}{n-2}} \gamma_{\frac{n}{n-1}} \right] + \left[ \gamma_{\frac{2}{1}}\gamma_{\frac{3}{2}} \ldots \gamma_{\frac{n-1}{n-2}} \frac{e_{n+1}^*}{\hat{e}_{n-1}^*} \right] \right)
$$

$$
= \hat{e}_1^* \lim_{n\to\infty} \left( 1 + \gamma_{\frac{2}{1}} + \gamma_{\frac{2}{1}}\gamma_{\frac{3}{2}} + \cdots + \left[ \gamma_{\frac{2}{1}}\gamma_{\frac{3}{2}} \ldots \gamma_{\frac{n-1}{n-2}} \frac{\hat{e}_n^*}{\hat{e}_{n-1}^*} \right] + \left[ \gamma_{\frac{2}{1}}\gamma_{\frac{3}{2}} \ldots \gamma_{\frac{n-1}{n-2}} \frac{e_{n+1}^*}{\hat{e}_{n-1}^*} \right] \right)
$$

$$
\leq \hat{e}_1^* \lim_{n\to\infty} \left( 1 + \gamma_{\frac{2}{1}} + \gamma_{\frac{2}{1}}^2 + \cdots + \gamma_{\frac{2}{1}}^{n-2} + \left[ \gamma_{\frac{2}{1}}^{n-2} \frac{\hat{e}_n^*}{\hat{e}_{n-1}^*} \right] + \left[ \gamma_{\frac{2}{1}}^{n-2} \frac{e_{n+1}^*}{\hat{e}_{n-1}^*} \right] \right)
$$

$$
= \left[ \hat{e}_1^* \lim_{n\to\infty} \left( 1 + \gamma_{\frac{2}{1}} + \gamma_{\frac{2}{1}}^2 + \cdots + \gamma_{\frac{2}{1}}^{n-2} \right) \right] + \left[ \hat{e}_1^* \lim_{n\to\infty} \left( \gamma_{\frac{2}{1}}^{n-2} \frac{(\hat{e}_n^* + e_{n+1}^*)}{\hat{e}_{n-1}^*} \right) \right]. \tag{32}
$$

*The first term in equation 32 forms a geometric series, and the second term in equation 32 is zero as $n$ goes to infinity, because $\hat{e}_1^*, \hat{e}_{n-1}^*, \hat{e}_n^*, e_{n+1}^*$ are bounded by construction and $\hat{e}_{n-1}^* > 0$ by Assumption 2. Hence,*

$$
|p(x,t) - \hat{p}(x,t)| \leq \hat{e}_1^* \left( \frac{1}{1 - \gamma_{\frac{2}{1}}} \right). \tag{33}
$$

## A.6 Proof of Theorem 2

**Proof 5** *We omit the time variable $t$ in this proof for readability. By Definition 1, the maximum approximation error $\max_x |e_1(x, \cdot)| := e_1^*$. Using the relations of $\hat{e}_1 = e_1 - e_2, \hat{e}_1^* \leq e_1^* + e_2^*$, the error bound in Theorem 1 can be upper-bounded by*

$$
e_B = \hat{e}_1^* \left( \frac{1}{1 - \hat{e}_2^*/\hat{e}_1^*} \right) \leq (e_1^* + e_2^*) \left( \frac{1}{1 - \hat{e}_2^*/\hat{e}_1^*} \right). \tag{34}
$$

*Hence, the gap between $e_B$ and the maximum approximation error $e_1^*$ is upper-bounded by*

$$
e_B - e_1^* \leq e_1^* \left( \frac{1}{1 - \hat{e}_2^*/\hat{e}_1^*} - 1 \right) + e_2^* \left( \frac{1}{1 - \hat{e}_2^*/\hat{e}_1^*} \right). \tag{35}
$$

*Now suppose $\hat{e}_1$ approximates $e_1$ sufficiently well such that $e_2(x,t) = e_1(x,t) - \hat{e}_1(x,t) := \delta(x,t)$, where $\delta(x,t)$ denotes a sufficiently small function for all $(x,t) \in \Omega$. Furthermore, suppose $\hat{e}_2$ approximates $e_2$ sufficiently well such that $\hat{e}_2(x,t) \to e_2(x,t) = \delta(x,t)$ for all $(x,t) \in \Omega$. Define $\delta^* := \max_x |\delta(x, \cdot)|$, then $\hat{e}_2^* \to \delta^*$, and $\delta^* \to 0$ as $\delta(x,t) \to 0$ for all $(x,t) \in \Omega$. Consequently, the RHS of equation 35, at the limit, becomes*

$$
\lim_{\hat{e}_2^* \to \delta^*, \delta^* \to 0} \left[ e_1^* \left( \frac{1}{1 - \hat{e}_2^*/\hat{e}_1^*} - 1 \right) + e_2^* \left( \frac{1}{1 - \hat{e}_2^*/\hat{e}_1^*} \right) \right]
$$

$$
= \lim_{\delta^* \to 0} \left[ e_1^* \left( \frac{1}{1 - \delta^*/\hat{e}_1^*} - 1 \right) + \delta^* \left( \frac{1}{1 - \delta^*/\hat{e}_1^*} \right) \right] = \delta^* \tag{36}
$$

*Lastly, for every $\epsilon \in (0, \infty)$, take $\delta^*$ to be smaller than $\epsilon$, then the proof is completed.*

## A.7 Proof of Corollary 1

**Proof 6** *The proof is a natural extension to that of theorem 1. Assume $m > 1$ be a finite integer. By Definition 1, we have*

$$p(x,t) - \hat{p}(x,t) = \lim_{n \to \infty} \sum_{i=1}^{n} \hat{e}_i(x,t) + e_{n+1}(x,t)$$

$$= \sum_{i=1}^{m-1} \hat{e}_i(x,t) + \lim_{n \to \infty} \Big( \sum_{i=m}^{n} \hat{e}_i(x,t) + e_{n+1}(x,t) \Big)$$

$$\implies p(x,t) - \hat{p}(x,t) - \sum_{i=1}^{m-1} \hat{e}_i(x,t) = \lim_{n \to \infty} \Big( \sum_{i=m}^{n} \hat{e}_i(x,t) + e_{n+1}(x,t) \Big)$$

$$\implies |p(x,t) - \hat{p}(x,t) - \sum_{i=1}^{m-1} \hat{e}_i(x,t)| = |\lim_{n \to \infty} \Big( \sum_{i=m}^{n} \hat{e}_i(x,t) + e_{n+1}(x,t) \Big)|$$

$$\implies |p(x,t) - \hat{p}(x,t) - \sum_{i=1}^{m-1} \hat{e}_i(x,t)| \leq \lim_{n \to \infty} \sum_{i=m}^{n} \hat{e}_i^*(x,t) + e_{n+1}^*(x,t)$$

$$\leq \hat{e}_m^*(t)\Big(1 + \frac{\hat{e}_{m+1}^*(t)}{\hat{e}_m^*(t)} + \frac{\hat{e}_{m+2}^*(t)}{\hat{e}_m^*(t)} + \cdots + \frac{\hat{e}_n^*(t)}{\hat{e}_m^*(t)} + \frac{e_{n+1}^*(t)}{\hat{e}_m^*(t)}\Big)$$

$$= \lim_{n \to \infty} \hat{e}_m^*\Big(1 + \gamma_{\frac{m+1}{m}} + \gamma_{\frac{m+1}{m}}\gamma_{\frac{m+2}{m+1}} + \ldots + \gamma_{\frac{m+1}{m}}\gamma_{\frac{m+2}{m+1}}\ldots\gamma_{\frac{n}{n-1}} \frac{e_{n+1}^*}{\hat{e}_n^*}\Big) \qquad (37)$$

*Under the same condition in lemma 3, but now impose on $\alpha_m(t)$ and $\alpha_{m+1}(t)$ such that $0 < \alpha_m(t) < 1$ and $0 < \alpha_{m+1}(t) < 1 - \alpha_m(t), \alpha_{m+1}(t)(1 + \alpha_{m+1}(t)) < \alpha_m^2(t)$. Then $\gamma_{\frac{m+1}{m}}(t) < 1$ is greater than all the other $\gamma_{\frac{m+2}{m+1}}(t), \gamma_{\frac{m+3}{m+2}}(t), \ldots$. Thus, equation 37 is bounded by*

$$|p(x,t) - \hat{p}(x,t) - \sum_{i=1}^{m-1} \hat{e}_i(x,t)| \leq \lim_{n \to \infty} \hat{e}_m^*(t)\Big(1 + \gamma_{\frac{m+1}{m}} + \gamma_{\frac{m+1}{m}}^2 + \cdots + \gamma_{\frac{m+1}{m}}^{n-1} + \gamma_{\frac{m+1}{m}}^{n-1} \frac{e_{n+1}^*}{\hat{e}_n^*}\Big)$$

$$= \Big[\hat{e}_m^*(t) \lim_{n \to \infty} \Big(1 + \gamma_{\frac{m+1}{m}} + \gamma_{\frac{m+1}{m}}^2 + \cdots + \gamma_{\frac{m+1}{m}}^{n-1}\Big)\Big] + \Big[\hat{e}_m^*(t) \lim_{n \to \infty} \alpha_{n-1}(t)\gamma_{\frac{m+1}{m}}^{n-1}(t)\Big]. \qquad (38)$$

*Since $\hat{e}_m^*(t)$ is bounded, $\gamma_{\frac{m+1}{m}} < 1$, and $\exists \alpha_{n-1} \leq \alpha_{m+1} < 1$, the first term in equation 38 forms a geometric series, and the second term goes to zero. Hence .equation 38 becomes*

$$|p(x,t) - \hat{p}(x,t) - \sum_{i=1}^{m-1} \hat{e}_i(x,t)| \leq \hat{e}_m^*(t)\Big(\frac{1}{1 - \gamma_{\frac{m+1}{m}}(t)}\Big) \implies$$

$$p(x,t) - \hat{p}(x,t) \in \Big[ \sum_{i=1}^{m-1} \hat{e}_i(x,t) - \hat{e}_m^*(t)\Big(\frac{1}{1 - \gamma_{\frac{m+1}{m}}(t)}\Big), \sum_{i=1}^{m-1} \hat{e}_i(x,t) + \hat{e}_m^*(t)\Big(\frac{1}{1 - \gamma_{\frac{m+1}{m}}(t)}\Big)\Big].$$

$$(39)$$

*Now take $n = m + 1$, then the proof is completed.*

## A.8 Proof of Corollary 2

**Proof 7** *For every $t \in \mathbb{R}_{\geq 0}$, let $0 < \alpha_1(t) < 1$. Suppose there exists a "virtual" $\hat{e}_2(x,t)$ such that $\hat{e}_2(x,t) = e_2(x,t)$ for all $(x,t) \in \Omega$ ; this implies that the third error $e_3(x,t)$ is zero. Hence, the series in equation 9 becomes finite*

$$|p(x,t) - \hat{p}(x,t)| \leq \hat{e}_1^*(t) + \hat{e}_2^*(t) + 0$$

$$= \hat{e}_1^*(t)\Big(1 + \gamma_{\frac{2}{1}}(t)\Big). \qquad (40)$$

*By the virtual $\hat{e}_2 = e_2$, and the relation $e_2^* = \alpha_1 \hat{e}_1^*$, we have*

$$\max_x |\hat{e}_2(x,t)| = \max_x |e_2(x,t)|$$
$$\implies \hat{e}_2^*(t) = e_2^* = \alpha_1(t)\hat{e}_1^*(t)$$
$$\implies \gamma_{\frac{2}{1}}(t) = \alpha_1(t). \tag{41}$$

*Combined $\gamma_{\frac{2}{1}} = \alpha_1$ with equation 40, we prove that*

$$|p(x,t) - \hat{p}(x,t)| \leq \hat{e}_1^*\Big(1 + \gamma_{\frac{2}{1}}(t)\Big)$$
$$= \hat{e}_1^*(t)\Big(1 + \alpha_1(t)\Big)$$
$$< \hat{e}_1^*(t)(1 + 1) = 2\hat{e}_1^*(t). \tag{42}$$

*It is clear that $e_S(t)$ is not arbitrary tight because of the constant 2.*

### A.9 PROOF OF PROPOSITION 1

**Proof 8** *Let $x \in \mathbb{R}^n$. By (Mishra & Molinaro, 2023, theorem 2.6), we know*

$$\varepsilon_G := \|e_1 - \hat{e}_1\|_{W^{1,q}} \leq C_{pde}\varepsilon_T + C_{pde}C_{quad}^{\frac{1}{q}}N^{\frac{-\beta}{q}}, \tag{43}$$

*where $C_{pde} > 0$ are the stability estimates of the first error PDE associated with the $W^{1,q}$ norm, $q \geq 2$, and $C_{quad}, \beta > 0$ are the constants according to the quadrature sampling points. For expression simplicity, denote $e_2 := e_1 - \hat{e}_1$. Since $e_1(x,t)$ and $\hat{e}_1(x,t)$ are bounded, we know there exists a universal embedding constant $C_{embed}$ (Mizuguchi et al., 2017) such that*

$$|e_2(x,t)| \leq C_{embed}\|e_2(x,t)\|_{W^{1,q}}. \tag{44}$$

*Hence, we have*

$$|e_2(x,t)| \leq C_{embed}\Big(C_{pde}\varepsilon_T + C_{pde}C_{quad}^{\frac{1}{q}}N^{\frac{-\beta}{q}}\Big). \tag{45}$$

*Using the definition of $\alpha_1(t) := \frac{\max_x |e_2(x,t)|}{\hat{e}_1^*(t)}$, we obtain*

$$\alpha_1(t) \leq \frac{\max_x |e_2(x,t)|}{\min_t \hat{e}_1^*(t)}$$
$$\leq \frac{1}{\min_t \hat{e}_1^*(t)}\Big[C_{embed}\Big(C_{pde}\varepsilon_T + C_{pde}C_{quad}^{\frac{1}{q}}N^{\frac{-\beta}{q}}\Big)\Big]. \tag{46}$$

## B  ADDITIONAL RESULTS OF NUMERICAL EXPERIMENTS

Here, we report training details and additional results of the numerical experiments. The baseline training scheme is done by randomly selecting space-time points at every training epoch. The other training scheme employs adaptive sampling and residual gradient loss suggested by (Lu et al., 2021) and (Yu et al., 2022). Adaptive sampling exploits the infinite training data property of physics-informed learning by automatically adding the space-time points whose residual values are large. Residual gradient loss is an additional physics-informed loss term that regularizes the change of residual with respect to space and time; it has been shown to stabilize and accelerate the training. We consider this regularization because the residual of $\hat{p}$, i.e. $\mathcal{D}[\hat{p}]$, is used as inputs to the subsequent training of $\hat{e}_1$. For completeness, we implement a normalized loss function based on equation 8 to train $\hat{e}_i$ for all $i \geq 0$:

$$\mathcal{L} = w_0 \mathcal{L}_0 + w_r \mathcal{L}_r + w_{\nabla r} \mathcal{L}_{\nabla r}, \quad \mathcal{N} = \max_{x_k \in X'} |e_i(x_k, 0)|$$

$$\mathcal{L}_0 = \frac{1}{N_0} \sum_j^{N_0} \|\frac{\hat{e}_i(x_j, 0) - e_i(x_j, 0)}{\mathcal{N}}\|_2^2, \quad \mathcal{L}_r = \frac{\text{Vol}(T)}{N_r} \sum_j^{N_r} \|\frac{\mathcal{D}[\hat{e}_i(x_j, t_j)] + r_i(x_j, t_j)}{\mathcal{N}}\|_2^2,$$

$$\mathcal{L}_{\nabla r} = \frac{\text{Vol}(T)}{N_r} \sum_j^{N_r} \|\nabla\Big(\frac{\mathcal{D}[\hat{e}_i(x_j, t_j)] + r_i(x_j, t_j)}{\mathcal{N}}\Big)\|_2^2, \tag{47}$$

where $\text{Vol}(T)$ is the duration of the time interval, $\mathcal{L}_{\nabla r}$ is the loss term of residual gradient, and $\mathcal{N}$ is a normalization constant. The baseline training has no regularization, i.e., $w_{\nabla r} = 0$. Both training schemes use Adam optimizer with initial learning rate $10^{-3}$ and exponentially decay learning rate.

### B.1  1D LINEAR SDE

We considered an 1D system (Ornstein-Uhlenbech process) $dx = -0.2xdt + \sqrt{0.4}dw$. Suppose the state is at $x^-$ at $t_{-1}$, then the analytical solution of $p(x, t)$ is $p(x, t) = \sqrt{\frac{0.2}{0.4\pi(1-e^{-0.4t})}} \exp\Big(-\frac{0.2(x-x^- e^{-0.2t})^2}{0.4(1-e^{-0.4t})}\Big)$. To avoid the initial distribution of a delta function $\delta(x - x^-)$, the initial distribution $p_0(x) = p(x, t = 1; x_{-1} = 1)$ is used. In this experiment, the input domain is: $x \in [-6, 6], t \in [1, 3]$. $\hat{p}(x, t)$ and $\hat{e}_1(x, t)$ are 2 hidden layers and 32 neurons MLPs using Softplus activation. Both neural networks initialize the weights using kaiming_normal_ and 0.01 bias. The baseline training scheme is used, i.e., randomly selected $N_0 = 500, N_r = 500$ space-time points are sampled at each epoch. The maximum training epochs for both $\hat{p}, \hat{e}_1$ are 2k. The weights of the loss function in equation 47 are $w_0 = 1, w_r = 1$ and $w_{\nabla r} = 0$. Training loss of $\hat{p}(x, t)$ and $\hat{e}_1(x, t)$ are shown in Fig. 5a. The artificial $\hat{e}_2(x, t)$ constructed by perturbing the true $e_2(x, t)$ is shown in Fig. 5b.

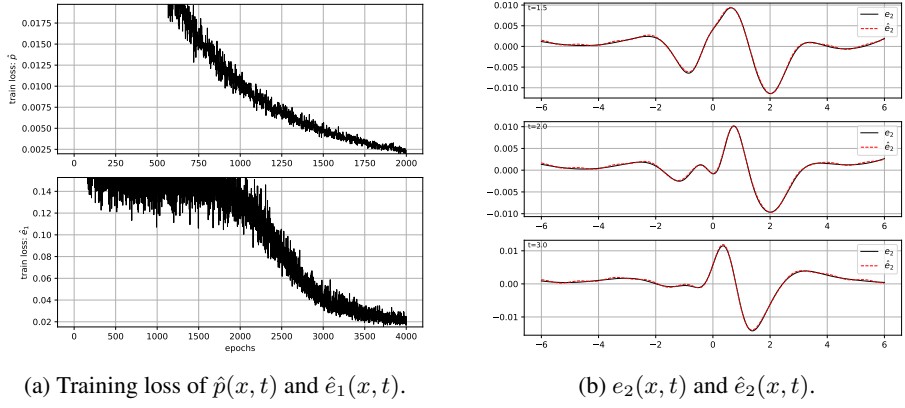

(a) Training loss of $\hat{p}(x, t)$ and $\hat{e}_1(x, t)$.       (b) $e_2(x, t)$ and $\hat{e}_2(x, t)$.

Figure 5: Training loss and synthesized $\hat{e}_2(x, t)$.

## B.2 1D Nonlinear SDE

Firstly, the "true" PDF $p(x, t)$ is obtained by extensive Monte-Carlo simulation of the SDE at some time instances using Euler Scheme; $\Delta t = 0.0005$ s, $\Delta x = 0.06$, and $10^9$ samples. This Monte-Carlo simulation took 100 hours on the MacBook Pro machine. The small time step and large samples are necessary to create accurate probability densities. Secondly, the result in Fig. 3(a)-(d) is obtained from $\hat{p}(x, t)$ using a 3 hidden layers 50 neurons Softplus activation MLP, and $\hat{e}_1(x, t)$ using a 6 hidden layers 50 neurons Softplus activation MLP. Both neural networks initialize the weights using kaiming_normal_ and 0.01 bias. The training scheme employs adaptive sampling and residual gradient loss, i.e., $w_0 = w_r = w_{\nabla r} = 1$. At the beginning of training, $N_0 = 1000, N_r = 1000$ space-time points are sampled from a uniform distribution. During training, 5 additional initial samples and 5 residual samples are added every 100 epochs. The maximum epochs for training $\hat{p}$ and $\hat{e}_1$ are 15000 and 25000, respectively. Figure 6a and Fig. 6b show the space-time samples (as blue dots) used during training. Figure 6c plots the training loss of $\hat{p}(x, t)$ and $\hat{e}_1(x, t)$; periodic spikes exist due to the adaptive sampling.

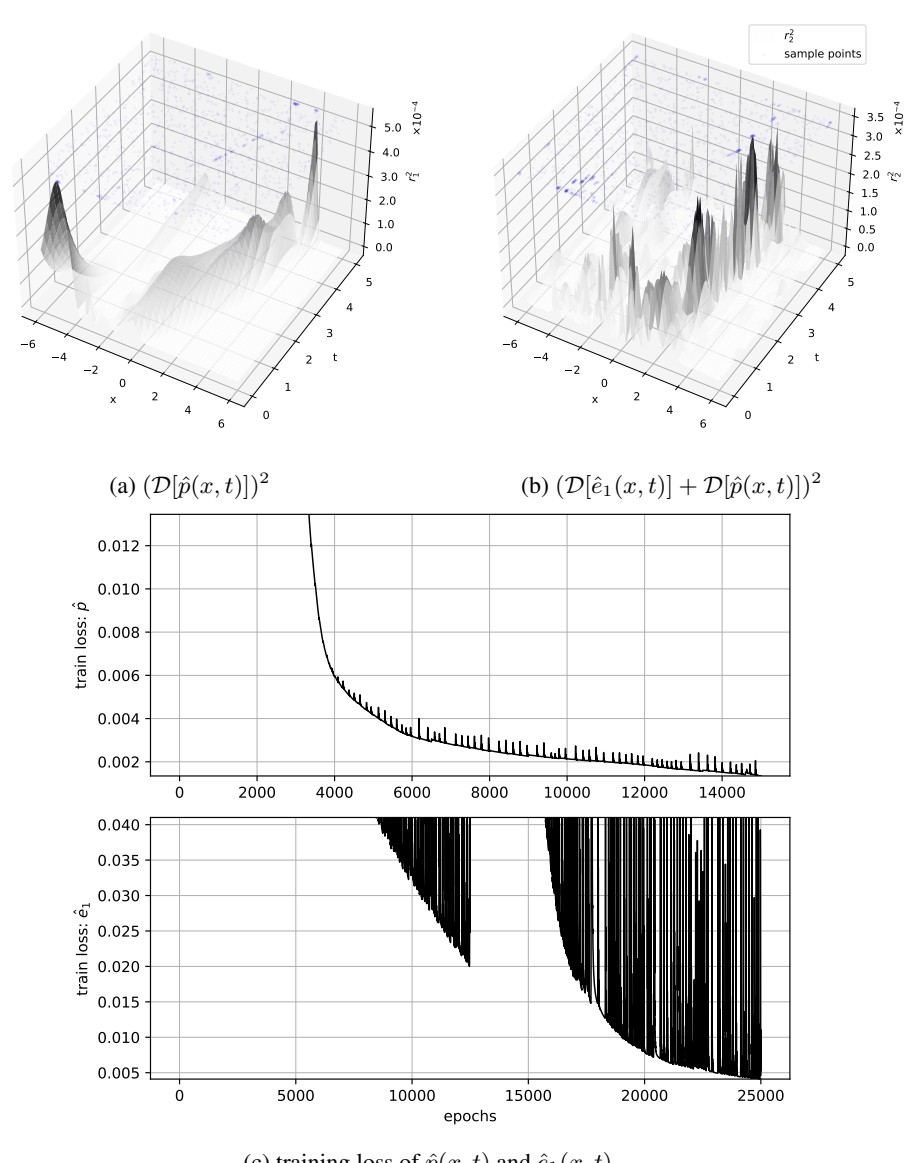

(a) $(\mathcal{D}[\hat{p}(x, t)])^2$          (b) $(\mathcal{D}[\hat{e}_1(x, t)] + \mathcal{D}[\hat{p}(x, t)])^2$

(c) training loss of $\hat{p}(x, t)$ and $\hat{e}_1(x, t)$.

Figure 6: Residuals and training loss of $\hat{p}(x, t)$ and $\hat{e}_1(x, t)$.

The results in Fig. 3(e)-(f) are obtained by training on the Linux desktop with GPU. $\hat{p}(x,t)$ is a 5 hidden layer 50 neurons MLP using GeLU activation for the hidden layers and Softplus activation for the final output (to ensure non-negative probability density). $\hat{e}_1(x,t)$ is a 5 hidden layer 50 neurons MLP using GeLU activation for the hidden layers. Both neural networks initialize the weights using kaiming_normal_ and 0.01 bias. The adaptive sampling and residual gradient loss are employed ($w_0 = w_r = w_{\nabla_r} = 1$). At the beginning of training $\hat{p}(x,t)$, $N_0 = 500$ and $N_r = 600$ space-time points are sampled uniformly, together with a deterministic set of 40 initial points and 1600 residual points from a uniform grid. One additional initial point and one residual point are added during training of $\hat{p}(x,t)$. At the beginning of training $\hat{e}_1(x,t)$, $N_0 = 500$ and $N_r = 1000$ space-time points are sampled uniformly, together with a deterministic set of 40 initial points and 1600 residual points from a uniform grid. One additional initial point and ten residual points are added during training of $\hat{e}_1(x,t)$. Both neural networks have maximum 50000 training epochs. The maximum training time of $\hat{p}(x,t)$ is 778 seconds; the maximum training time of $\hat{e}_1(x,t)$ is 49643 seconds. Below from Fig. 7 to Fig. 12, we report the first order temporal error bound results of all the six trials, each using different random seed.

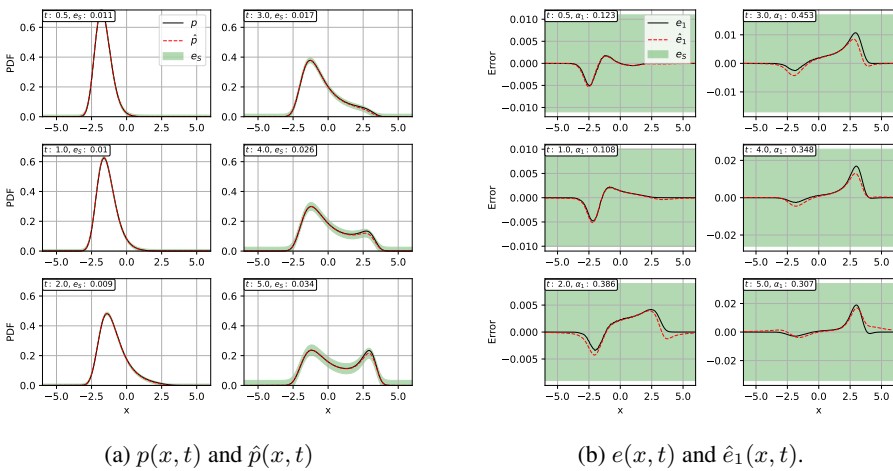

(a) $p(x,t)$ and $\hat{p}(x,t)$

(b) $e(x,t)$ and $\hat{e}_1(x,t)$.

Figure 7: First order temporal error bounds of GeLU neural networks, random seed= 0.

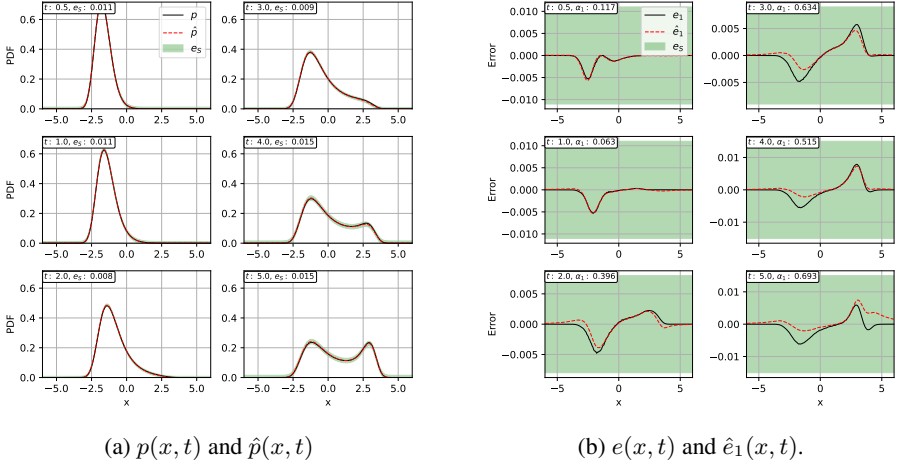

(a) $p(x,t)$ and $\hat{p}(x,t)$

(b) $e(x,t)$ and $\hat{e}_1(x,t)$.

Figure 8: First order temporal error bounds of GeLU neural networks, random seed= 1.

Lastly, we report the first order temporal error bound results if the neural networks are trained without residual gradient regularization. In this training setting, the weights of the loss are set to $w_0 = 1, w_r = 2$ and $w_{\nabla_r} = 0$, and the maximum training epochs are also 50000 for both $\hat{p}$ and $\hat{e}_1$. Figure 13 compares the training results of using adaptive sampling and residual gradient regularization (top row) vs only using adaptive sampling (bottom row). The former has

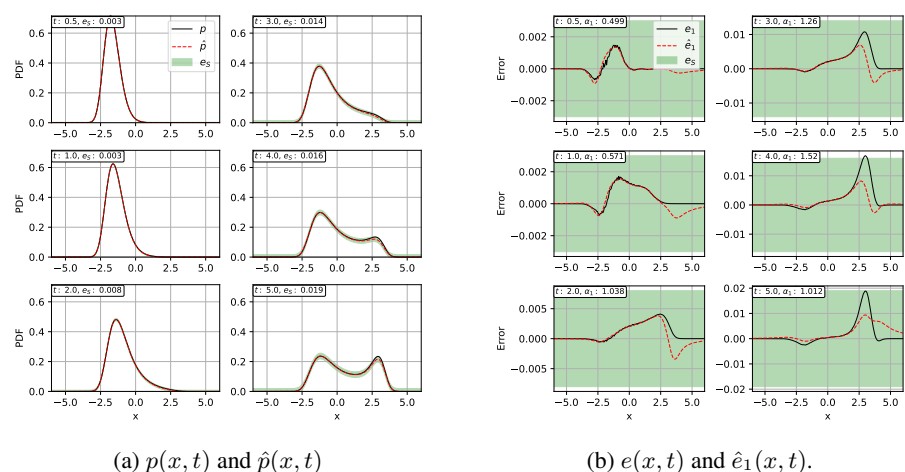

(a) $p(x,t)$ and $\hat{p}(x,t)$

(b) $e(x,t)$ and $\hat{e}_1(x,t)$.

Figure 9: First order temporal error bounds of GeLU neural networks, random seed= 2.

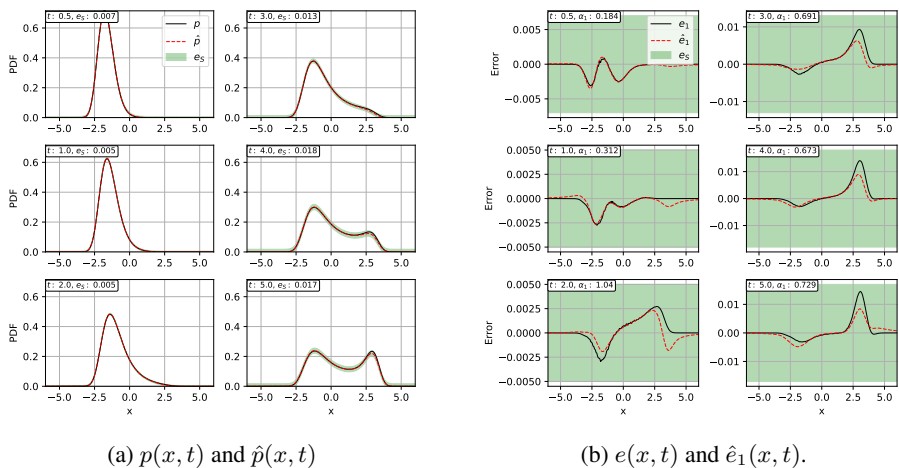

(a) $p(x,t)$ and $\hat{p}(x,t)$

(b) $e(x,t)$ and $\hat{e}_1(x,t)$.

Figure 10: First order temporal error bounds of GeLU neural networks, random seed= 3.

$\max_t e_S(t) = 0.115, \max_t \alpha_1(t) = 0.6$, and the latter has $\max_t e_S(t) = 0.158, \alpha_1(t) = 0.917$. In terms of learning time, the latter is faster; it takes 225 seconds for $\hat{p}(x,t)$ and 866 seconds for $\hat{e}_1(x,t)$, while training using both residual gradient regularization and adaptive sampling is slower: $\hat{p}(x,t)$ for 715 seconds and $\hat{e}_1(x,t)$ for 3868 seconds.

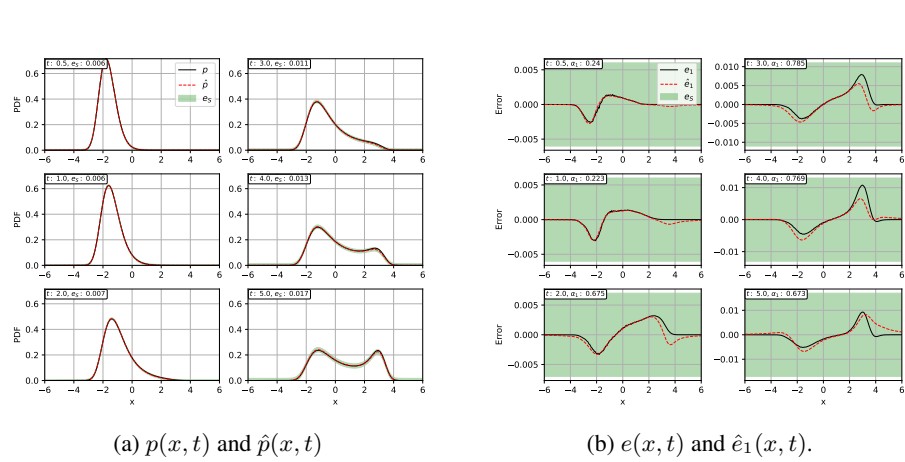

(a) $p(x,t)$ and $\hat{p}(x,t)$

(b) $e(x,t)$ and $\hat{e}_1(x,t)$.

Figure 11: First order temporal error bounds of GeLU neural networks, random seed= 4.

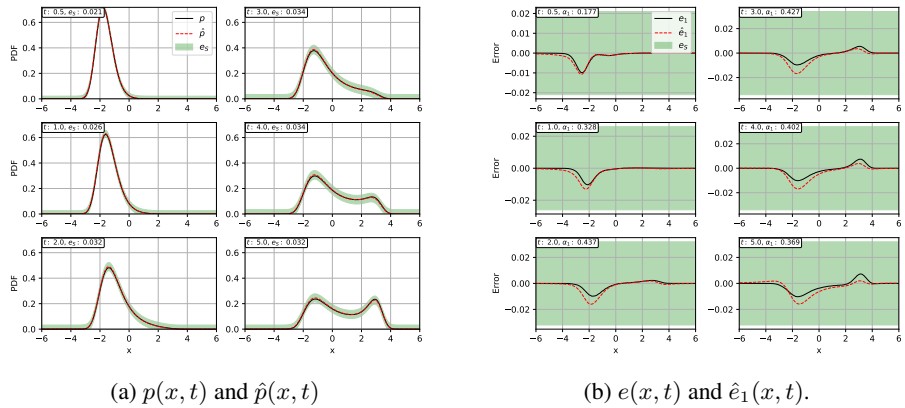

(a) $p(x,t)$ and $\hat{p}(x,t)$

(b) $e(x,t)$ and $\hat{e}_1(x,t)$.

Figure 12: First order temporal error bounds of GeLU neural networks, random seed= 5.

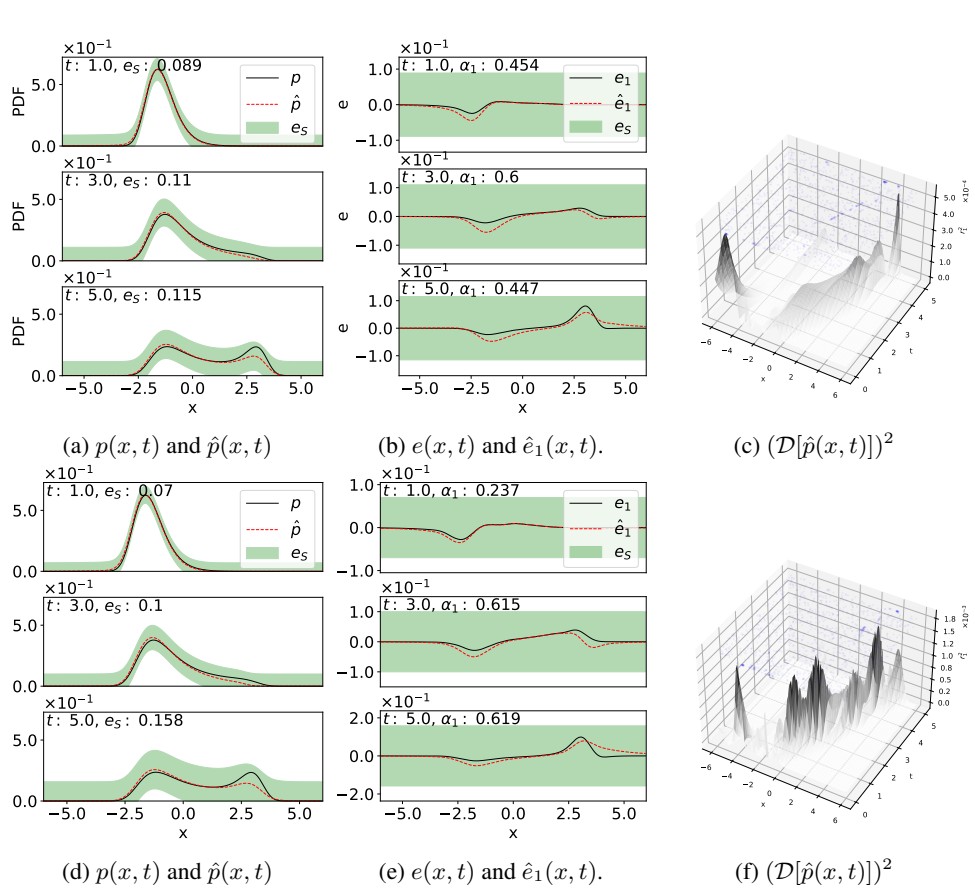

(a) $p(x,t)$ and $\hat{p}(x,t)$     (b) $e(x,t)$ and $\hat{e}_1(x,t)$.     (c) $(\mathcal{D}[\hat{p}(x,t)])^2$

(d) $p(x,t)$ and $\hat{p}(x,t)$     (e) $e(x,t)$ and $\hat{e}_1(x,t)$.     (f) $(\mathcal{D}[\hat{p}(x,t)])^2$

Figure 13: Comparison of different training schemes. Top: adaptive sampling and residual gradient regularization. Bottom: only adaptive sampling.

## B.3 1D SDE with State-dependent Noise

We considered a 1D SDE with state-dependent noise (also known as geometric brownian motion) $dx = (ax)dt + (bx)dw$, where $x \in \mathbb{R}$ is the state. The associated FP-PDE is $\frac{\partial p}{\partial t} + \frac{\partial [axp]}{\partial x} - \frac{1}{2}b^2\frac{\partial^2 [x^2 p]}{\partial x^2} = 0$. By (Shreve et al., 2004), a special analytical solution of the FP-PDE exists if $x > 0$: $p(x,t) = 1/(bx\sqrt{2\pi t})\exp(-(\log\frac{x}{x^-} - \nu t)^2/(2b^2 t))$, where $\nu = a - \frac{b^2}{2}$, and $\delta(x - x^-)$ is the initial delta distribution. Similar to 1D linear SDE, we let $t_0 = 1$ such that $p_0(x)$ is not a delta function (boundedness assumption). The input domain is $x \in [90, 110], t \in [1, 6]$; the parameters are $(a, b, x_0) = (0.002, 0.01, 100)$. $\hat{p}(x,t)$ is a 5 hidden layers 32 neurons MLP using Softplus activation. $\hat{e}_1(x,t)$ is a 5 hidden layers 64 neurons MLP using Softplus activation. Since the state domain is large $x \in [90, 100]$, $\hat{p}(x,t)$ and $\hat{e}_1(x,t)$ transform the state input to $\bar{x} = (x - 100)/100$, then pass $\bar{x}$ to the first hidden layer. Both neural networks initialize the weights using kaiming_normal_ and 0.01 bias. The adaptive sampling and residual gradient loss is employed during training, i.e., $w_0 = w_r = w_{\nabla r} = 1$. At the beginning of training, $N_0 = 1000$ initial points are sampled, half of which are sampled from the initial Gaussian distribution, the others are sampled from uniform distribution; $N_r = 1000$ residual space-time points are sampled from uniform distribution. During training, one residual space-time point is added every 100 epochs. Figs. 14a and 14b plot the solution, error, and the neural network approximations; $\hat{p}(x,t)$ and $\hat{e}_1(x,t)$ are trained with 0.0045 loss for 30 seconds and 0.005 loss for 598 seconds, respectively. The first order temporal error bound at $t = \{2.0, 4.0, 6.0\}$ is illustrated in the solution and error spaces in Fig. 14c and Fig. 14d, respectively. Again, $e_S(t)$ successfully constructs a tight temporal error bound if $\alpha_1(t)$ condition is satisfied. Figure 15a plots the training residuals of the neural network at specific time instances; By Definition 1, we desire $\mathcal{D}[\hat{e}_1(x,t)] \to -\mathcal{D}[\hat{p}(x,t)]$. Due to adaptive sampling, periodic spikes are present in Fig. 15b as well.

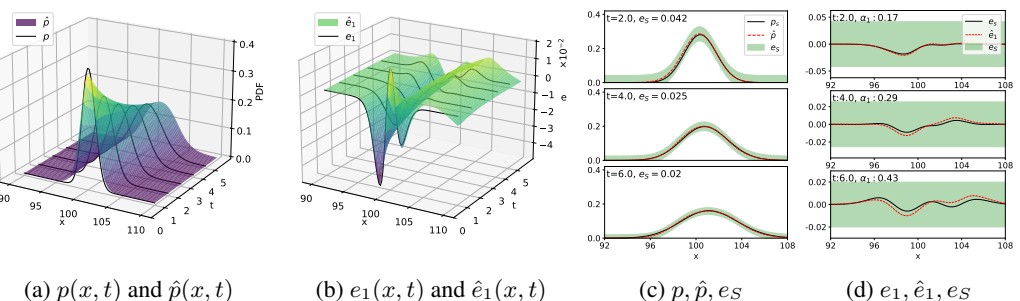

(a) $p(x,t)$ and $\hat{p}(x,t)$     (b) $e_1(x,t)$ and $\hat{e}_1(x,t)$     (c) $p, \hat{p}, e_S$     (d) $e_1, \hat{e}_1, e_S$

Figure 14: First order temporal error bound of the 1D SDE with state-dependent noise.

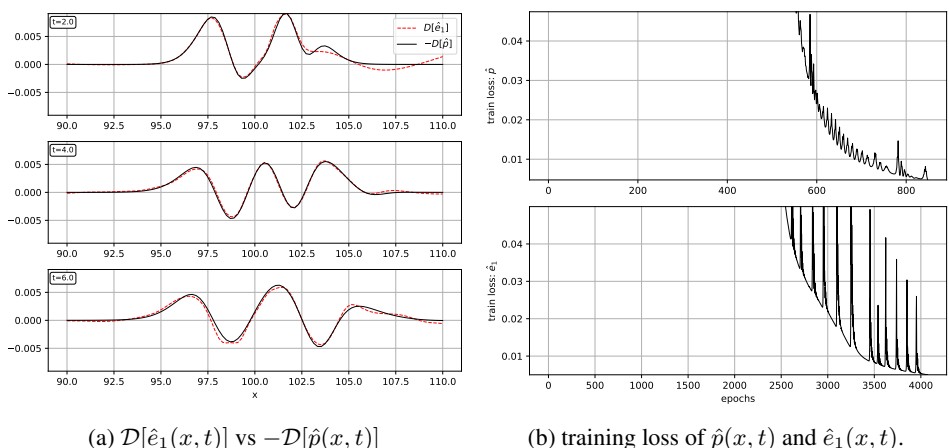

(a) $\mathcal{D}[\hat{e}_1(x,t)]$ vs $-\mathcal{D}[\hat{p}(x,t)]$       (b) training loss of $\hat{p}(x,t)$ and $\hat{e}_1(x,t)$.

Figure 15: Residuals and training loss of $\hat{p}(x,t)$ and $\hat{e}_1(x,t)$.

## B.4  NONLINEAR INVERTED PENDULUM

We considered an inverted pendulum system given by $dx = f(x)dt + Bdw$, where $x = [\theta, \dot{\theta}]^T \in \mathbb{R}^2$ is the state, $f(x) = [x_2, -\frac{g}{l}\sin(x_1)]^T$, $g$ is the gravity acceleration, $l$ is the length of the inverted pendulum, $B \in \mathbb{R}^{2\times2}$, and $dw \in \mathbb{R}^2$. The initial distribution is a multivariate Gaussian $x_0 \sim \mathcal{N}(\mu_0, \Sigma_0)$. In this experiments, the input domain is $x_1 \in [-3\pi, -3\pi], x_2 \in [-3\pi, -3\pi]$, and $t \in [0, 5]$. The parameters are $(g, l, B, \mu_0, \Sigma_0) = (9.8, 9.8, [0.5, 0.0; 0.0, 0.5], [0.5\pi, 0.0]^T, [0.5, 0.0; 0.0, 0.5])$. Similarly, $p(x, t)$ is obtained by Monte-Carlo simulation of the SDE at some time instances using Euler Scheme; $\Delta t = 0.01$ s, $\Delta x_1 = 0.3768, \Delta x_2 = 0.3768$, and $10^8$ samples. This Monte-Carlo simulation took 13 hours on the MacBook Pro machine. $\hat{p}(x, t)$ is a 5 hidden layers 32 neurons MLP using Softplus activation. $\hat{e}_1(x, t)$ is a 7 hidden layers 32 neurons MLP using Softplus activation. Both neural networks initialize the weights using kaiming_normal_ and 0.01 bias. Adaptive sampling and residual gradient is used during training, again, $w_0 = w_r = w_{\nabla r} = 1$. $N_0 = 500$ initial points and $N_r = 1500$ residual space-time points are sampled uniformly at the beginning of training. Additional 5 initial and 5 residual points are added every 100 epochs during training. Figure 16 plots the $p(x, t)$ in the first row, and the trained $\hat{p}(x, t)$ in the second row. The approximation error is plotted in the first row in Fig. 17, while the second row shows the first error approximation $\hat{e}_1(x, t)$. Fig. 18 shows a 3d surface plot of the absolute errors $|e(x, t)|$, which are upper-bounded by the surface of the $e_S(t)$. Figure 19 plots the training loss of $\hat{p}(x, t)$ and $\hat{e}_1(x, t)$.

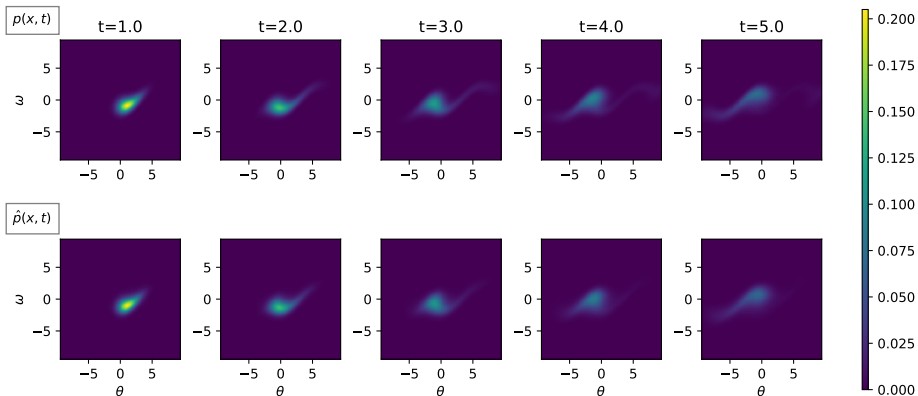

Figure 16: $p(x, t)$ and $\hat{p}(x, t)$ at $t = \{1.0, 2.0, 3.0, 4.0, 5.0\}$.

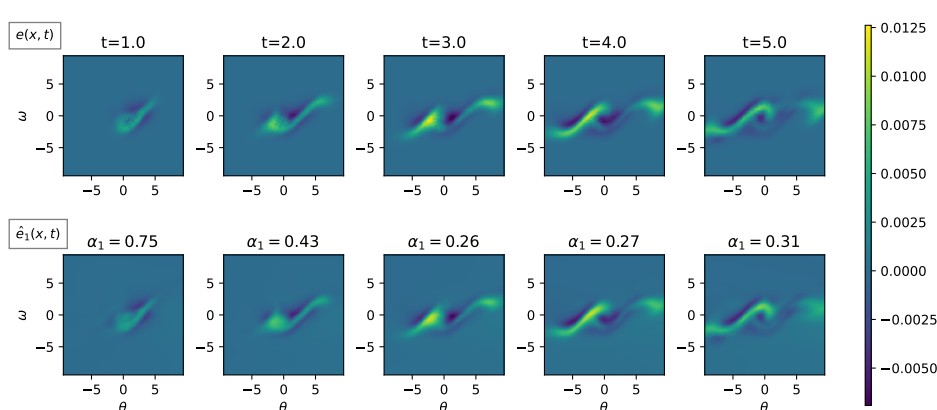

Figure 17: $e(x, t)$ and $\hat{e}_1(x, t)$ at $t = \{1.0, 2.0, 3.0, 4.0, 5.0\}$.

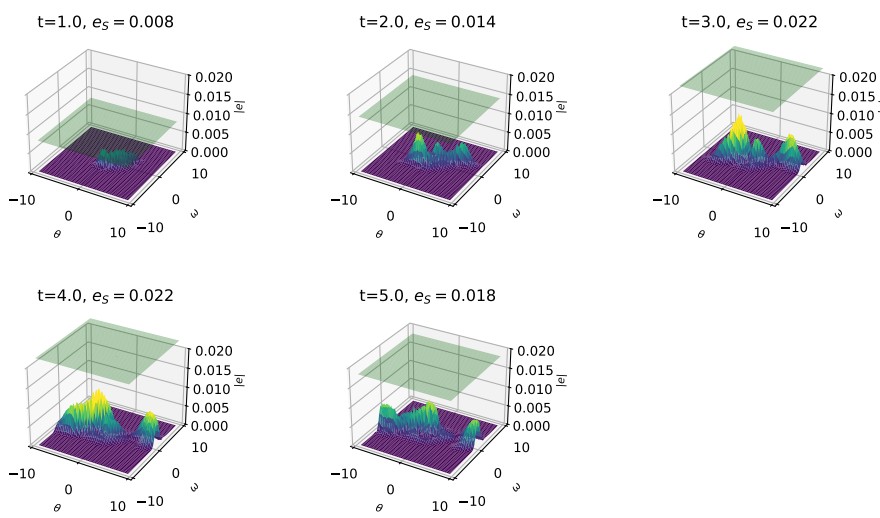

Figure 18: Absolute errors $|e(x, t)|$ and first order temporal error bounds $e_S(t)$ (illustrated as the green surface) at $t = \{1.0, 2.0, 3.0, 4.0, 5.0\}$.

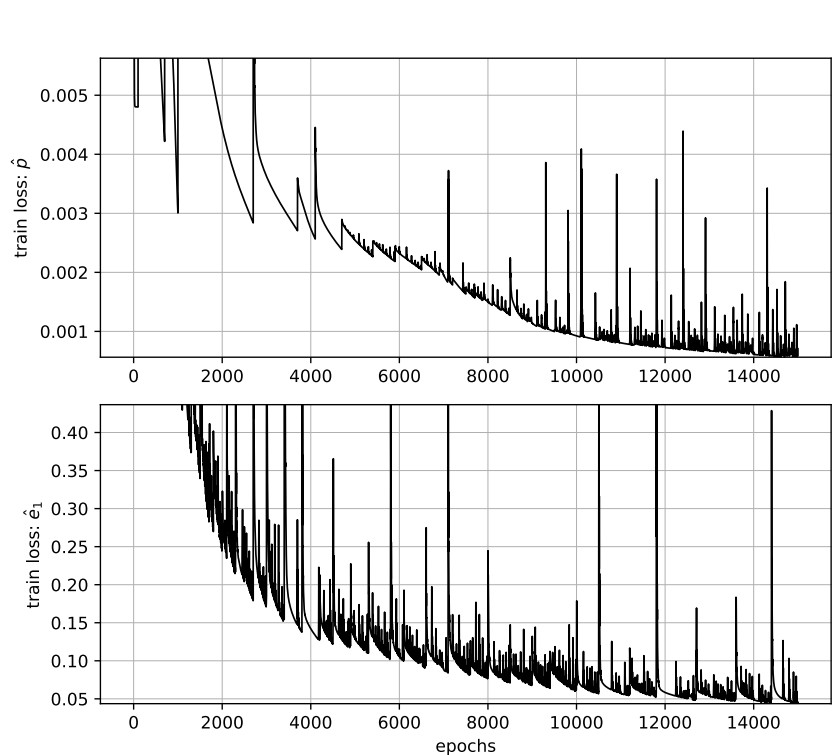

Figure 19: training loss of $\hat{p}(x, t)$ and $\hat{e}_1(x, t)$.

## B.5   1D HEAT EQUATION

We considered an one-dimensional heat equation $u_t - u_{xx} = 0$ with boundary condition, $u(\pm 1, t) = 0, \forall t$. Let $t_0 = 0$, and the initial distribution $u_0(x) = -\sin(\pi x)$. In this experiments, the input domain is $x \in [-1, 1], t \in [0, 1]$. For this particular problem, analytical solution exists: $u(x, t) = -\sin(\pi x) \exp^{-\pi^2 t}$, which allows us to validate the first order temporal error bound using trained $\hat{u}(x, t), \hat{e}_1(x, t)$. $\hat{u}(x, t)$ is a 3 hidden layers 64 neurons MLP using $\mathrm{Tanh}$ activation. $\hat{e}_1(x, t)$ is a 5 hidden layers 100 neurons MLP using $\mathrm{Tanh}$ activation. Both neural networks initialize the weights using $\mathrm{xavier\_uniform\_}$ and zero bias. The baseline training scheme ($w_0 = w_r = 1$) is used with $N_0 = 500, N_r = 500$ random samples at each epoch. Figure 21 show the residuals and training loss of the neural networks. Again, we desire $\mathcal{D}[\hat{e}_1(x, t)] \rightarrow -\mathcal{D}[\hat{u}(x, t)]$ for good training.

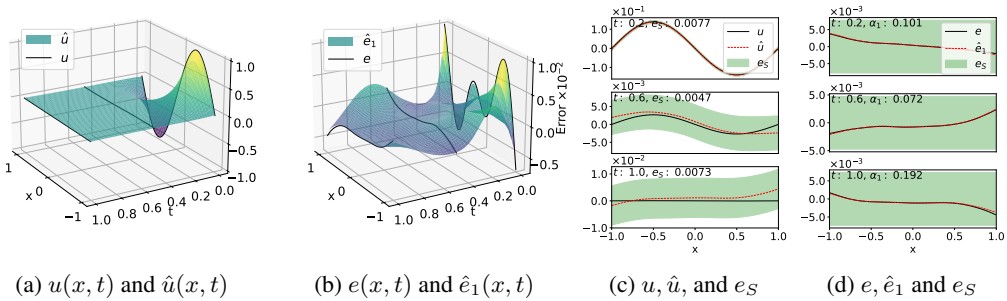

(a) $u(x,t)$ and $\hat{u}(x,t)$     (b) $e(x,t)$ and $\hat{e}_1(x,t)$     (c) $u, \hat{u}$, and $e_S$     (d) $e, \hat{e}_1$ and $e_S$

Figure 20: first order temporal error bound of 1D heat equation.

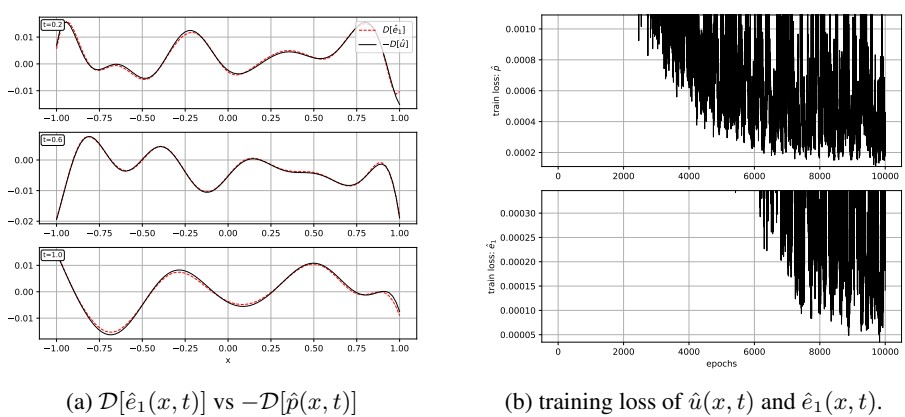

(a) $\mathcal{D}[\hat{e}_1(x,t)]$ vs $-\mathcal{D}[\hat{p}(x,t)]$        (b) training loss of $\hat{u}(x,t)$ and $\hat{e}_1(x,t)$.

Figure 21: Residuals and training loss of $\hat{u}(x,t)$ and $\hat{e}_1(x,t)$.

### B.6 HIGH-DIMENSIONAL ORNSTEIN-UHLENBECK

We considered the generalization of the 1D Ornstein-Uhlenbeck process to $n$-dimension with time-varying dynamics: $dx = (A_n(t)x)dt + B_n dw$, where $x, w \in R^n$, and the initial distribution is multi-variate Gaussian $p(x, 0) \sim \mathcal{N}(\mu_n, \Sigma_n)$. For this system, the probability density functions over time remains Gaussian $p(x, t) \sim \mathcal{N}(\mu_n(t), \Sigma_n(t))$, but there is no close-form solution to $\mu_n(t)$ and $\Sigma_n(t)$ in general (Särkkä & Solin, 2019). Here, we use Euler forward numerical integration (0.0001 seconds time step) to obtain the "true" PDF. The solution domain we tested is $\Omega = [-1, 1]^n \times [0, 1]$.

For the 3D OU, the dynamics is $A_3 = \begin{bmatrix} 0.3 & 0.0 & 0.0 \\ 0.0 & 0.3 & 0.0 \\ -0.1 & 0.0 & 0.3 \end{bmatrix}$, $B_3 = \text{diag}([0.05, 0.05, 0.05])$ and the intial distribution is $\mu_3 = [-0.2, 0.2, 0.0], \Sigma_3 = \text{diag}([0.1, 0.1, 0.1])$. For the 3D time-varying OU, the dynamics is $\tilde{A}_3(t) = \begin{bmatrix} 0.3 & 0.0 & 0.0 \\ 0.0 & 0.3 & 0.0 \\ -0.1 & 0.0 & 0.3 \end{bmatrix} + (e^{-t^3}) \begin{bmatrix} 0.0 & 0.5 & 0.0 \\ 0.0 & 0.0 & 0.5 \\ 0.0 & -0.3 & 0.0 \end{bmatrix}$ with the same noise coupling and initial distribution as the 3D OU. For the 7D OU, the dynamics is an almost diagonal $A_7 = \text{diag}([0.05, 0.05, 0.05, 0.05, 0.05, 0.05, 0.05])$ with $A_7[7, 1] = -0.01$, $B_7 = \text{diag}([0.05, 0.05, 0.05, 0.05, 0.05, 0.05, 0.05])$. The initial distribution is $\mu_7 = [0.0, 0.0, 0.0, 0.0, 0.0, 0.0, 0.0], \Sigma_7 = \text{diag}([0.12, 0.12, 0.12, 0.12, 0.12, 0.12, 0.12])$. For the 10D OU, the dynamics is an almost diagonal $A_{10} = \text{diag}([0.05, ...0.05])$ with $A_{10}[10, 1] = -0.01$, $B_{10} = \text{diag}([0.05, ..., 0.05])$. The initial distribution is zero mean $\mu_{10} = [0.0, ..., 0.0], \Sigma_{10} = \text{diag}([0.11, ..., 0.11])$. For the 10D time-varying OU, the dynamics is $\tilde{A}_{10}(t) = A_{10} + (e^{-t^3})\Delta A_{10}$, where $\Delta A_{10}$ is first initialing a zero 10 by 10 matrix, then setting $\Delta A_{10}[1, 2] = 0.1, \Delta A_{10}[2, 3] = 0.1$, and $\Delta A_{10}[10, 2] = -0.1$. The noise coupling is the same as the 10D OU, the initial distribution is $\tilde{\mu}_{10} = [-0.2, 0.1, 0.2, 0.05, -0.25, 0.22, 0.18, -0.12, 0.01, 0.04]$, and the covariance is the same as well. For all the experiments (3D-10D), we use the same neural networks: $\hat{p}(x, t)$ and $\hat{e}_1(x, t)$ are 5 hidden layers 32 neurons MLP using GeLU activation; the final output of $\hat{p}$ is passed into Softplus to ensure non-negative value. Both neural networks initialize the weights using kaiming_normal_ and 0.0 bias. The adaptive sampling is employed during training, i.e., $w_0 = w_r = 1$. At the beginning of training, $N_0 = N_r = 2000$ points are sampled for $\hat{p}$, and $N_0 = N_r = 300$ points are sampled for $\hat{e}_1$. Additional 40 samples are added for both trainings if the loss of the current epoch is smaller than 0.95 times the minimum loss. After training, we evaluate the results at uniform time instances $t = \{0.0, 0.2, 0.4, 0.6, 0.8, 1.0\}$. For each time instances, the evaluated state points are chosen by (1) a deterministic uniform grid ($50 \times 50 \times 50$) for the 3D cases, or (2) uniformly $10^7$ samples at random for the 7D and 10D cases. Figs. 22- 26 report (i) the first order temporal error bound $e_S(t)$ versus the maximum error $\max_x |e_1(x, t)|$ for all time (normalized by $\max_x |p(x, t)|$ as used in Table. 2), (ii) the condition $\alpha_1(t) < 1$, $\forall t \in T$, and (iii) the training history of $\hat{p}$ and $\hat{e}_1$. Lastly, Figs. 27 and 28 visualize the PDF $p(x, t)$, the PDF approximation $\hat{p}(x, t)$, the approximation error $e_1(x, t)$, and the first error approximation $\hat{e}_1(x, t)$ as 3D contour plots for the 3D Time-varying OU.

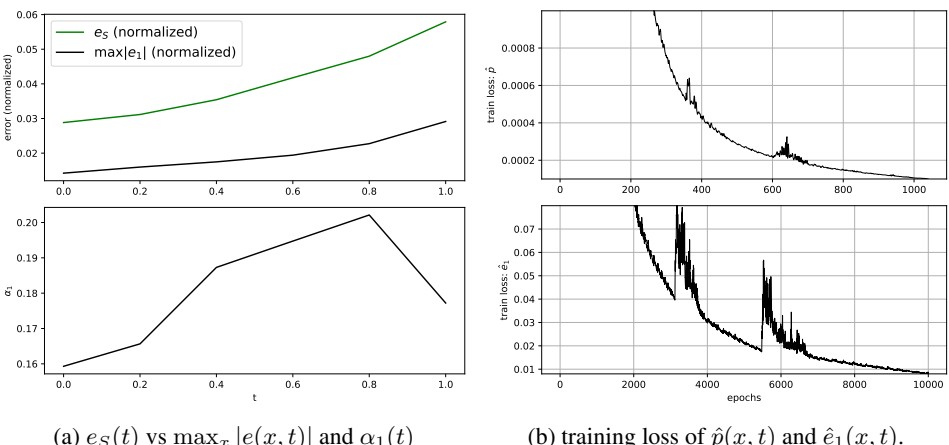

(a) $e_S(t)$ vs $\max_x |e(x, t)|$ and $\alpha_1(t)$

(b) training loss of $\hat{p}(x, t)$ and $\hat{e}_1(x, t)$.

Figure 22: Results of the first-order temporal error bound of 3D OU.

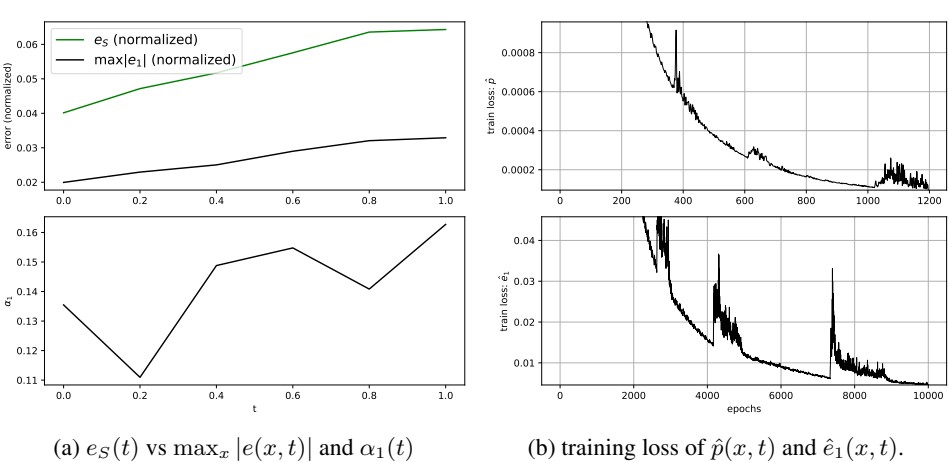

(a) $e_S(t)$ vs $\max_x |e(x,t)|$ and $\alpha_1(t)$      (b) training loss of $\hat{p}(x,t)$ and $\hat{e}_1(x,t)$.

Figure 23: Results of the first-order temporal error bound of 3D Time-varying OU.

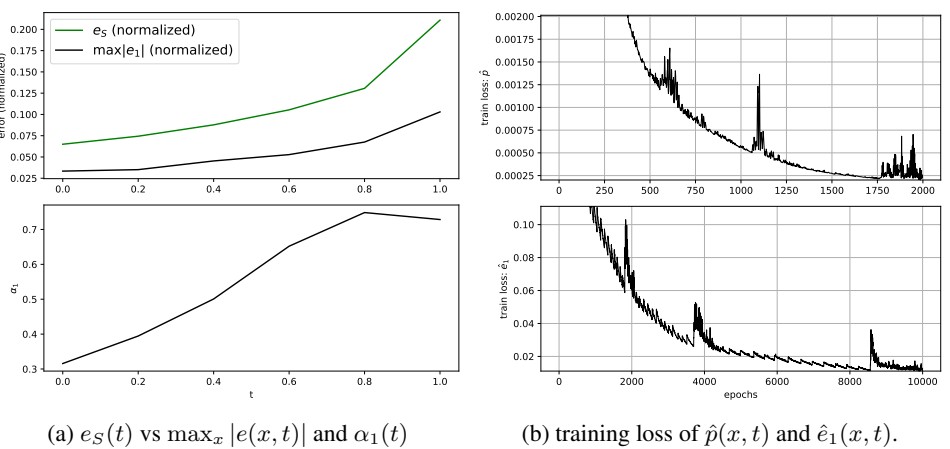

(a) $e_S(t)$ vs $\max_x |e(x,t)|$ and $\alpha_1(t)$      (b) training loss of $\hat{p}(x,t)$ and $\hat{e}_1(x,t)$.

Figure 24: Results of the first-order temporal error bound of 7D OU.

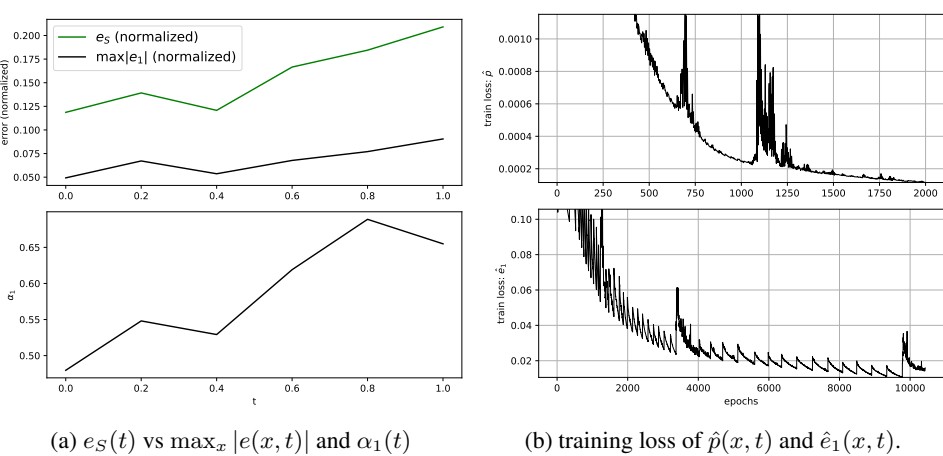

(a) $e_S(t)$ vs $\max_x |e(x,t)|$ and $\alpha_1(t)$      (b) training loss of $\hat{p}(x,t)$ and $\hat{e}_1(x,t)$.

Figure 25: Results of the first-order temporal error bound of 10D OU.

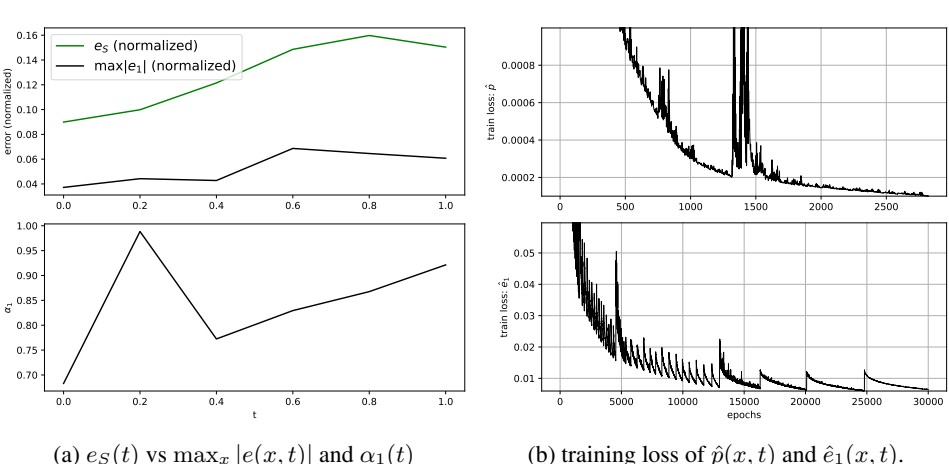

(a) $e_S(t)$ vs $\max_x |e(x,t)|$ and $\alpha_1(t)$

(b) training loss of $\hat{p}(x,t)$ and $\hat{e}_1(x,t)$.

Figure 26: Results of the first-order temporal error bound of 10D Time-varying OU.

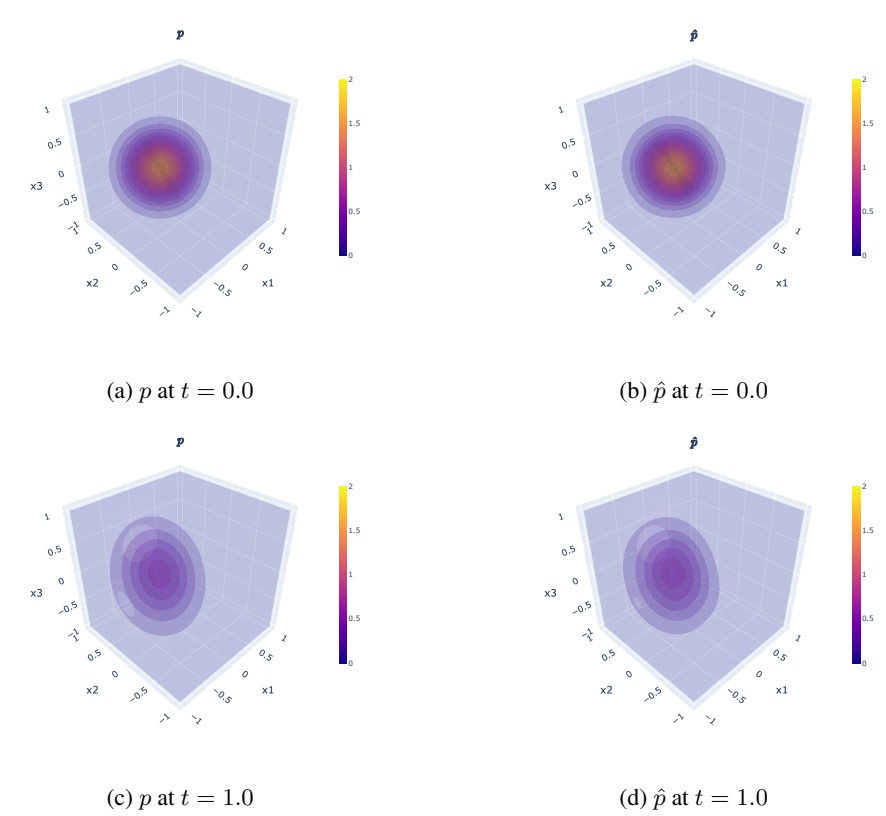

(a) $p$ at $t = 0.0$

(b) $\hat{p}$ at $t = 0.0$

(c) $p$ at $t = 1.0$

(d) $\hat{p}$ at $t = 1.0$

Figure 27: 3D Time-varying OU: PDF $p$ and the neural network approximation $\hat{p}$ for different $t$.

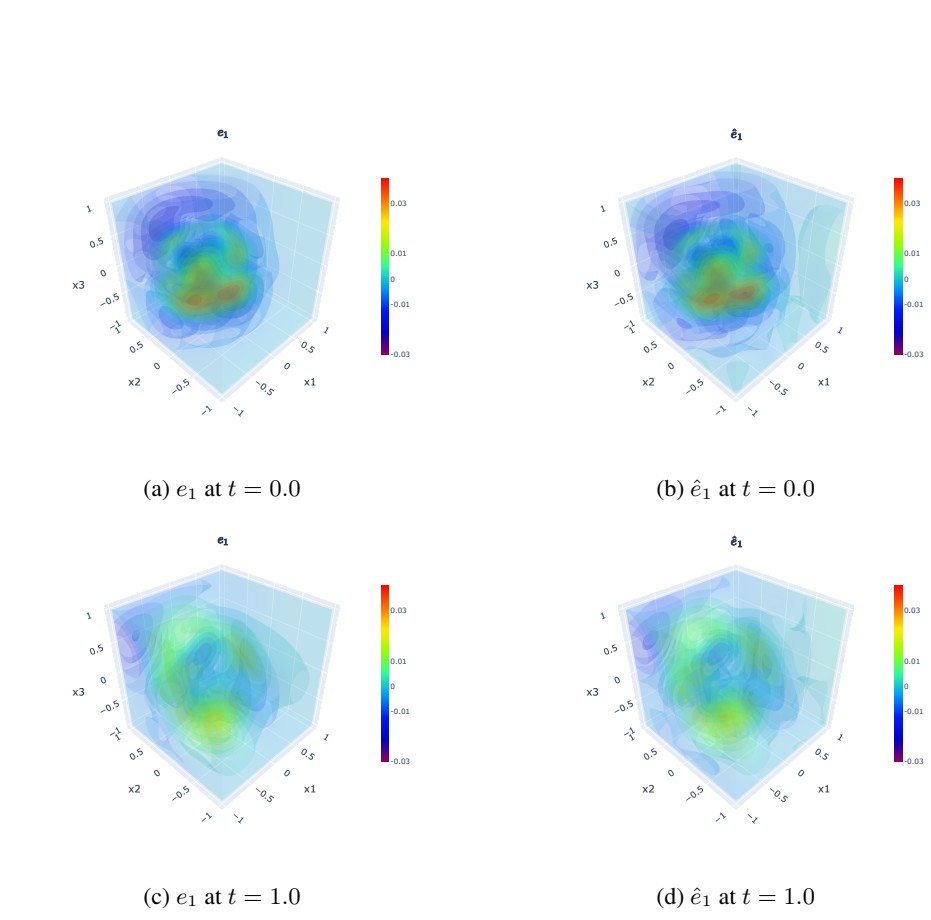

(a) $e_1$ at $t = 0.0$        (b) $\hat{e}_1$ at $t = 0.0$

(c) $e_1$ at $t = 1.0$        (d) $\hat{e}_1$ at $t = 1.0$

Figure 28: 3D Time-varying OU: approximation error $e_1$ and the neural network approximation $\hat{e}_1$ for different $t$.

