# OpenReview forum: "Error Bounds for Deep Learning-based Uncertainty Propagation in SDEs"
_ICLR.cc/2025/Conference — Submitted to ICLR 2025_

### Official Review · Reviewer_7jTR · 2024-10-23

**Soundness:** 3
**Presentation:** 2
**Contribution:** 3
**Rating:** 6
**Confidence:** 3

**Summary:**

Paper Summary:

Physics-informed neural networks (PINNs) can be used to approximate the solution probability density function (PDF) for stochastic differential equations (SDEs). While existing methodologies can achieve high levels of approximation accuracy, the absence of knowledge regarding the true solution renders it impossible to assess the quality of the approximated solution definitively. The authors’ main contribution is the establishment of a general framework for deriving an arbitrarily tight error bound that quantifies the discrepancy between the true solution and its approximation, utilizing two additional PINNs. Furthermore, this paper presents a more practical error bound that relies solely on one additional PINN, albeit at the expense of losing the property of arbitrary tightness.
The methodology for the more practical bound is shown on a set of systems for which a true solution can be derived.
The paper presents a novel method for estimating the accuracy of an approximated PDF. Although the approach does not yield an exact metric for assessing whether any approximated PDF is well-trained, it offers a way that can, though it does not necessarily guarantee, confirm the well-trained status of an approximated PDF. This bound allows users to assess and decide whether the approximation is acceptable based on their specific criteria.

Review Summary:

Overall, I really love the idea and think the novelty is "good". However, the presentation needs to improved and and the points flagged as "Major" below should be adressed in order to to be able to recommend acceptance.


EDIT: Raised to 6 after author response.

**Strengths:**

The authors present an (to the best of my knowledge) novel approach that leverages the linearity of the FP-PDE operator to enable a derivation of an error bound, thereby enhancing the reliability of solution PDF approximations.

The well-structures theoretical section provides a clear and sound proof that is thoroughly derived.

**Weaknesses:**

Major:

The authors position their method as a general framework applicable to the n-dimensional case. However, the experiments conducted primarily concentrate on the one-dimensional scenario, with the inclusion of a single two-dimensional case.

While it has been demonstrated, that an arbitrary tight second-order temporal error bound can be achieved, this is contingent upon specific conditions being met for α1 and α2. However, the paper only provides a method to verify α1, but not addresses α2, rendering the second-order temporal bound impractical for use.

Page 10, Figure 3g illustrates the fragility of the method with respect to the fulfillment of the necessary conditions (16a), highlighting that its effectiveness is highly sensitive to random initialization. This suggests that the method’s practicability may be constrained by the high variance of α1-values.


Minor:

Page 6, Remark 4 refers to equation 17, however equation 15 describes the relation ship between γ and α.

Page 10, Figure 3: Not all y-axes are defined.

**Questions:**

Major:

Can you demonstrate your method on a high-dimensional example, such as a 10-dimensional Ornstein-Uhlenbeck?

Since that the true solution is known for the experiments conducted, it would be beneficial to compare the second-order temporal error bound with the first-order temporal error bound presented in Table 2 on page 8. Given the paper’s assertion that arbitrary tightness can be achieved with the second-order temporal error bound, it would be interesting to observe this in the experimental results. Additionally, analyzing the corresponding α2-values will provide insights into whether the necessary conditions (16b) are met.

Including a table (or described as additive in Table 2, page 8) that presents the variance of α1-values would provide valuable insights into whether the method can be reliably used to estimate the accuracy of the approximated PDF.

Minor:

Page 4, Definition 1: The paper claims, that the PINN that approximates ei is governed by the recursive PDE D[ei(x,t)]+D[êi−1(x,t)]=0. However, I could not directly verify this assertion from the preceding equations. Instead, my educated guess would yield a different outcome D[ei(x,t)]+Σk=1i−1D[êk(x,t)]=−D[^p(x,t)]. It would be valuable for unfamiliar readers if the authors would clarify the origin of the abbreviated formulation.

"1D Nonlinear SDE (GPU, seed0)." It would be beneficial to specify whether the results for "1D Nonlinear SDE" represent average values over multiple seeds. Furthermore, clarification is needed on whether any seeds were excluded due to α1-values failing to satisfy the necessary condition (16a) at the end of training, which would render the first-order temporal error bound uninterpretable. (Page 10, Figure 3g)

---

> ### Author Response · Authors · 2024-11-20
>
> Thank you for the constructive feedback and comments.
>
> **Q. Major 1 / Weakness Major 1**
> * We agree with this weakness of the original submission, and we thank you for your constructive suggestion on how to address it.  Accordingly, we have run additional experiments for 3 to 10 dimensional Ornstein-Uhlenback systems. The results are in the official comment above, which we will also add to the paper.  Note that the framework is able to scale to 10D OU systems with a large but satisfying value for $\alpha_1$ and a small variance on $\alpha_1$. Nevertheless, the error bound remains small even for the 10D case.
> * For a detailed discussion on the scalability of the first-order error bound for high-dimension and large-scale systems, we kindly refer the reviewer to our response to Reviewer PJyn, Q3.
>
> **Q. Major 2 / Weakness Major 2**
> * **Q.Major.2:** Please note that the original submission includes a comparison of the second-order and first-order temporal error bounds via the 1D Linear SDE. We compared the magnitudes of these error bounds in Fig. 2a, showing the the second-order error bound is tighter than the first-order one for all time. Additionally, Fig. 2b-d reported the values of $\alpha_1$ and $\alpha_2$, showing the satisfaction of the sufficient conditions of $\alpha_1$ and $\alpha_2$ (Eq. 16a-c).
> * **W.Major.2:** We acknowledge that our 2nd-order error bound, while theoretically feasible, lacks an explicit method for checking its conditions ($\alpha_2$). However, we emphasize that this work is the first to quantify the worst-case approximation error of PINNs. The primary focus of this paper is to establish the theoretical foundation and demonstrate its feasibility. To address practical concerns, Corollary 2 outlines an approach to construct error bounds with reasonable tightness. Developing a fully practical computational framework based on these theoretical foundations is an ongoing effort.
>
> **Q. Major 3 / Weakness Major 3**
> * We agree that $\alpha_1$ may initially fluctuate due to random initialization and training of neural networks. The same Fig. 3g, however, also shows that $\alpha_1$ reliably decreases once the loss value becomes sufficiently small ($<0.1$) and as more training data are added. In fact, this behavior is guaranteed by Proposition 1.
> * As suggested by the reviewer, we have now reported the variance of $\alpha_1$ for additional experiments. We also reported the variance of $\alpha_1$ over all the multiple trials of the 1D Nonlinear SDE on GPU; specifically, see footnote of the Table 2 update above.
> Observe that max value of $\alpha_1$ is $<1$ and variance remains small. We will similarly expand Table 2 in the paper.
> * That said, we reiterate that the aim of this work is to lay down the theoretical framework and demonstrate feasibility using existing training methods. Ongoing investigations focus on reliable implementations, e.g., joint training of the PDF PINN $\hat{p}$ and error PINNs.
>
> **Q. Minor 1**
> * Thank you for catching that. That is a mistake we made when we were simplifying the notations in Definition 1. The correct recursive PDE (Eq.8) should be ($i\geq 1$):
> $$\mathcal{D}[e_i]+\sum_{j=1}^{i} \mathcal{D}[\hat{e}_{j-1}(x,t)]=0.$$
>
> * Below, we provide a complete clarification.
>
>     * Firstly, the intuition of the recursive PDE in Definition 1 was briefly provided in page 4, line 197: "Note that FP-PDE operator $\mathcal{D}$ is a linear operator; hence, by applying it to $e(x, t)$, we obtain: $\mathcal{D}[e]=\mathcal{D}[p-\hat{p}]=\mathcal{D}[p]-\mathcal{D}[\hat{p}].$ As $\mathcal{D}[p]=0$ (by Eq.4), .... we can define the governing PDE of $e(x,t)$ as $$\mathcal{D}[e(x,t)]+\mathcal{D}[\hat{p}(x,t)]=0,\quad\text{ subject to }\quad e(x,0)=p_0(x)-\hat{p}(x,0).\quad\text{ Eq.7"}$$
>
>     * Secondly, denote $e:=e_1$ as the first error and initialize $e_0:= p$ and $\hat{e}_0=\hat{p}$. Then, Eq.7 becomes Definition 1 for $i=1$: $$\mathcal{D}[e_1]+\mathcal{D}[\hat{e}_0]=0\text{, subject to }e_1(x,0)=e_0(x,0)-\hat{e}_0(x,0).$$ For $i=2$, we define $e_2:=e_1-\hat{e}_1$ and obtain $\mathcal{D}[e_2]=\mathcal{D}[e_1]-\mathcal{D}[\hat{e}_1]$. Since $\hat{e}_1\neq e_1$, we have $$\mathcal{D}[\hat{e}_1]+\mathcal{D}[\hat{e}_0]:=r_1\neq 0.$$ Hence, the PDE for $i=2$ is $$\mathcal{D}[e_2]=\mathcal{D}[e_1]-\mathcal{D}[\hat{e}_1] = (-\mathcal{D}[\hat{e}_0])-(-\mathcal{D}[\hat{e}_0]+r_1) = -r_1 \implies \mathcal{D}[e_2]+r_1 :=   \mathcal{D}[e_2] + \sum _{j=1} ^2 \mathcal{D}[ \hat{e} _{j-1}] = 0.$$ The derivation recursively follows for $i>2$.
>
>     We will add a similar clarification in the paper.
>
> **Q. Minor 2**
> * Each row in Table 2 is the result of one random seed. We will state this in the revised paper.
> * For the 1D Nonlinear SDE system, we performed 6 random trials on GPU (Figs. 7-12); one trial (seed2) failed to train $\hat{e}_1$, i.e., did not satisfy Eq.16a for some $t$ (as seen in Fig. 9). We will also add this note to the paper.

---

> > ### Comment · Reviewer_7jTR · 2024-11-22
> > **Thank you for your reply**
> >
> > Thank you for your reply. I raised my score to 6 after rebuttal.

---

### Official Review · Reviewer_sySV · 2024-10-28

**Soundness:** 3
**Presentation:** 2
**Contribution:** 2
**Rating:** 5
**Confidence:** 4

**Summary:**

The paper addresses error bounds for deep learning-based uncertainty propagation in stochastic differential equations by leveraging physics-informed neural networks (PINNs) to approximate probability density functions governed by the Fokker-Planck equations. The authors propose a method to construct error bounds for these approximations. Specifically, they introduce both a recursive error approximation method and a practical single-layer error bound, demonstrating the efficacy of these bounds in safety-critical systems. Experiments validate the theoretical contributions, showing the method’s effectiveness in bounding errors for various SDEs.

**Strengths:**

1. The paper focuses on a practical aspect of uncertainty quantification in deep learning applications for SDEs, which is valuable for many domains. The proposed recursive error function approach fills a gap by offering a way to bound approximation errors tightly.
2. The authors provide rigorous proofs, including convergence criteria and bounding error functionals, which are clearly laid out and generalized to linear PDEs. Theorems on the sufficiency of using only two PINNs for error bounding and proofs of tightness for error bounds add theoretical robustness to the claims.
3. The experimental results are broad, covering multiple SDEs and boundary conditions, which show the practical relevance of the proposed bounds. The results validate that the first-order temporal error bound covers approximation errors while remaining relatively tight, particularly in terms of normalized error gap.

**Weaknesses:**

1. The recursive approach, while theoretically sound, might pose practical challenges in computational efficiency and model training times for higher-dimensional SDEs. Although the authors suggest that two PINNs suffice in most cases, a deeper discussion of the computational overhead in high dimension cases would be necessary.
2. The practicality of checking conditions such as those outlined for $\alpha_1$ and $\alpha_2$ during training remains unclear. While conditions are theoretically justified, their feasibility in large-scale implementations with multiple variables is not discussed in depth.

**Questions:**

1. Could you provide further insights into the computational costs associated with the recursive error approximation, especially in higher dimensions?
2. Are there specific use cases where the single-layer error bound would fail to capture significant error characteristics, or where it might be insufficient compared to the recursive bound?
3. Could you elaborate on the practicality of verifying conditions for \alpha_1\ and \alpha_2 during training, particularly in large-scale applications?
4. There are many recent works on using PINNs to approximate probability distributions governed by the Fokker-Planck equations. In particular, the committor function. For example, Solving high dimensional committor problems by (1) Neural Networks (https://arxiv.org/abs/1802.10275) (2) Neural Network based-Symbolic Regression (https://arxiv.org/abs/2306.12268) (3) PINN + adaptive sampling for solving committors (https://arxiv.org/abs/2404.06206). A discussion of related work would enhance the impact of the work.

---

> ### Author Response · Authors · 2024-11-20
>
> **Q1**
> * As the reviewer noted, the sufficient conditions in Eq.16 imply that the computational cost of the second-order recursive error bound (Theorem 1) is significantly higher than that of the first-order error bound, *regardless of the system's dimension*. This is expected, as the second-order recursive error bound can theoretically achieve arbitrary tightness. In practice, however, the first-order error bound is more effective, particularly for high-dimensional systems. In fact, per reviewers' suggestions, we conducted additional experiments demonstrating scalability to larger systems, up to 10 dimensions. The results are provided in the official comment above, which will also be added to the paper.
>
> **Q2**
> * The main difference between the first-order error bound (Corollary 2) and the recursive error bound of 2nd order (Theorem 1) is their tightness. As long as the sufficient conditions (Corollary 2 and Eq.16, respectively) are satisfied, both of them will capture the error characteristics.
> * In Fig. 2, we empirically validate that the former is at most 2 times the size of the latter.
> * However, both share one common issue: PINNs could suffer to learn highly oscillating functions. This issue can be addressed by using regularization schemes (discussed in Appendix B) to regulate the training of PDF neural network, which in turns regulates the unknown error function to be learned. These techniques were employed in our case studies, enabling scalability to 10D systems while maintaining small error bounds.
>
> **Q3**
> * This paper establishes a method of checking for $\alpha_1$ condition during training (Proposition 1); checking for $\alpha_2$ remains an open problem.
>
>     **Checking $\alpha_1<1$ in large-scale and high-dimensional systems:** Based on the constants $C_{pde},C_{quad},C_{embed}$, this condition becomes more difficult to satisfy (requiring smaller training loss of the first error neural network) as the state dimension and the size of the state and time domain increases. We kindly refer the reviewer to the response to *Reviewer PJyn, Q3* for a detailed discussion.
>
>     Also, we would like to emphasize that this is the first work to quantify the worst-case approximation error of PINN. Hence, the focus of this paper is the  foundation and its theoretical feasibility. Corollary 2 proposes a practical approach to construct such error bound with reasonable tightness. Developing a practical computational framework on these theoretical foundations is an extension that we're currently working on.
>
> **Q4**
> * Thank you for these references. We were not aware of them, and we will include a discussion on them in the related work section.
> * *Short Summary:* Recent works (Khoo et al., 2019; Song et al., 2023; Lin \& Ren, 2024) present numerical methods for approximating committor functions, which should satisfy Fokker-Planck equations. Compared to our work, the main distinction is that their approaches do not provide quantified error bounds, which is a key contribution of our work. Additionally, their approximate solution is a transition probability between two regions, whereas ours is the evolution of the probability density function over the entire domain.
>
> **Weakness 1**
> * We acknowledge that our 2nd-order error bounds are currently theoretical and further investigation is required to make them practical. However, these theoretical foundations lay the groundwork for future research aimed at developing practical computational frameworks.
>
> * Also, for clarification, we did not show two PINNs suffice for *most* cases. Instead, we proved that two PINNs (under certain conditions) are sufficient to bound the error with arbitrary tightness for **all** cases (Theorem 1 and 2).
>
> **Weakness 2**
> * We have run additional experiments for 3 to 10 dimensional Ornstein-Uhlenback systems (suggested by Reviewer 7jTR). The results, provided in the official comments above, demonstrate that our framework effectively scales to 10D systems with small error bounds. We will add these experiments to the paper.

---

> > ### Author Response · Authors · 2024-12-01
> >
> > We hope the explanations provided above have thoroughly addressed the reviewer’s questions and concerns. Please let us know if you need any further clarifications.  Again, thank you for your time and thoughtful consideration.

---

### Official Review · Reviewer_PJyn · 2024-10-30

**Soundness:** 3
**Presentation:** 3
**Contribution:** 2
**Rating:** 6
**Confidence:** 2

**Summary:**

The authors consider an approach for bounding the errors for physics informed neural networks (PINNs) approximating the solution to the Fokker-Planck equation. The Fokker-Planck equation satisfies a linear operator and establishes a connection between an underlying stochastic differential equation (SDE) and the parameters of the operator. PINNs consider applying a boundary condition with a differential operator through automatic differentiation to impose the solution of a PDE. The authors consider a new PDE that operators on the space of errors and analyze the behavior of this PDE. They break this PDE into a set of $n$ subsolutions whose sum is equal to the true error of the PINN approximate to the PDE. The proof techniques only rely on the underling PDE being linear and apply to other linear PDEs. Finally, the authors benchmark on some numerical experiments. They consider experiments on how the proposed bounds compare to the true empirical errors.

**Strengths:**

Methods for estimating errors of PDEs are important for making the neural network based PDE solvers applicable in real world scenarios.

The proof techniques are intuitive and can be applied to other settings with other linear PDEs.

The bounds also appear to be fairly tight from the numerical simulations.

**Weaknesses:**

The bounds are somewhat cumbersome to compute in practice due to the necessity of fitting two functions.

In the case of fitting a single function, there is an decrease in the tightness of the bound, but it becomes more feasible to compute.

It's unclear what the feasibility of training the error networks is.

**Questions:**

How does the method perform in higher dimensional cases? For example, there's a connection between the flows given by the Fokker Planck equation and generative modeling. However, the support is over typically very high dimensional spaces in the generative modeling cases. How is the empirical performance in such cases? PINNs sometimes have trouble in high dimensional cases, so I'm curious about the application here.

The loss for the error network training appears to be much more volatile than the original PDE network error, is there a reason why this is the case? It seems to be more difficult to train than the original PDE approximation.

Are the $C_{PDE}, C_{embedding}, C_{quadrature}$ constants easy to compute? How does the different quadrature rule affect the constant, for example? Additional information on the interpretation or finding these would be helpful.

---

> ### Author Response · Authors · 2024-11-20
>
> **Q1**
> * Thank you for this comment. We have a two-part response:
>
>   1. Existing works have shown that PINNs can perform well in high-dimensional systems. For example, (Sirignano \& Spiliopoulos, 2018) demonstrated PINNs on a system with 200 dimensions.
>   2. To further address the reviewer's comment, we conducted additional experiments on 3- to 10-dimensional Ornstein-Uhlenbeck systems, which typically require numerical integration when the dynamics are time-varying (Sarkka \& Solin, 2019). The results are provided in the official comment above. They demonstrate that our method scales effectively to 10D systems while maintaining tight error bounds.
>
> **Q2**
> * This is a very good observation by the reviewer: as shown in Figs. 6, 15, and 19, the training loss of $\hat{e_1}$ seems more volatile than that of $\hat{p}$. This arise from the recursive term in the $\hat{e_1}$ loss function. According to Eq. 8 (for $i=1$), $\hat{e_1}$ is trained with the loss $\mathcal{L} = w_0 \mathcal{L_0} + w_r  \mathcal{L_r}$, where
>   $$ \mathcal{L_0}= \frac{1}{N_0} \sum_{j=1}^{N_0} \| e(x_j, 0) - \hat{e_1}(x_j, 0) \|^2 , \quad \mathcal{L_r}= \frac{1}{N_r}\sum_{j=1}^{N_r}\|\mathcal{D}[\hat{e_1}(x_j,t_j)]+\mathcal{D}[\hat{p}(x_j,t_j)]\|^2. $$
>   Training of $\hat{e}_1$ requires the term $\mathcal{D}[\hat{p}(x_j,t_j)]$ as inputs, which makes training less stable. To mitigate this, we regularize the training of $\hat{p}$ to make the subsequent training of $\hat{e}_1$ easier, as described in Appendix B. In addition, the large spikes observed in the $\hat{e}_1$ training loss are caused by the adaptive sampling approach, where new samples of large residuals are added to the training samples periodically; this is explained in Appendix B.2.
>
>   In this work, we focused on theoretical derivation of error bounds and demonstrated the feasibility of separately training the PDF and error neural networks using regularization and adaptive sampling. Our future work will focus on simpler, faster, and more reliable training methods. For example, the "joint training" approach mentioned in the conclusion is an area we're investigating now.
>
> **Q3**
> * In short, the values of $C_{pde}$ and $C_{embed}$ increase with the state dimension and the size of the solution domain, making the condition on $\alpha_1$ in Proposition 1 more challenging to satisfy. Below, we provide additional explanations and references:
>
>   * **Growth of $C_{pde}$:** Cauchy-Schwarz and Grönwall inequalities can be used to (over-)estimate $C_{pde}$ (Mishra \& Molinaro, 2023), which typically grows exponentially with $t$.
>
>   * **Effect of $C_{quad}$:** In Eq. 22, $C_{quad}$ is multiplied by $N^{-\beta}$, where $N$ is the number of quadrature points and $\beta > 0$ is the convergence rate. A sufficiently large $N$ can ensure $C_{quad}N^{-\beta} \ll 1$. For example:
>
>     * Using standard Gauss quadrature rules, $C_{quad}N^{-\beta}$ is proportional to $\frac{1}{2N!}$ (Stoer et al., 1980).
>
>     * With low-discrepancy sequences, $C_{quad}N^{-\beta}$ converges at a rate of $(\log(N))^n N^{-1}$ (Mishra \& Molinaro, 2023).
>
>   * **Domain dependency of $C_{embed}$:** The explicit values of $C_{embed}$ depend on the domain geometry and size, as studied in (Mizuguchi et al., 2017). For example:
>
>     * 2D square domain (area 1): $C_{embed} = 5.6119$.
>
>     * 2D triangular domain (area 0.433): $C_{embed} = 4.7971$.
>
>     * 3D cube domain (volume 1): $C_{embed} = 22.6274$.
>
>   To further clarify these points in the paper, we have updated Proposition 1 to a more general setting in the revised version.
>
> **Weakness 1-3**
> * Yes, we agree that it is challenging, in practice, to construct "arbitrary tight" error bounds using the second-order temporal error bound (Theorems 1 and 2).
> * However, we would like to emphasize that the focus of this paper is mainly on theoretical derivations of tight error bounds using finite number of PINNs, along with its sufficient conditions.
> * Further, to provide practical bounds, we also derive the first-order temporal error bound (Corollary 1) and its checking condition during training (Proposition 1).
> * We illustrated that constructing the practical bound using simple neural networks and existing training techniques is possible and in fact can scale to 10 dimensional systems.
> * With the theoretical foundation in place, our future work will be focused on developing more effective training methods for these error PINN networks.

---

> > ### Comment · Reviewer_PJyn · 2024-11-24
> >
> > Thank you for your responses, they helped my understanding of the paper.

---

### Official Review · Reviewer_tHQ1 · 2024-10-30

**Soundness:** 3
**Presentation:** 3
**Contribution:** 2
**Rating:** 3
**Confidence:** 3

**Summary:**

The paper studies developing data-driven error bounds for stochastic differential equation. The solution builds on the idea of constructing a recursion stack of excess errors made by a Physics-Informed Neural Network (PINN) on the solution of the Fokker-Plack equation. The trick that makes these error bounds feasible is that the error made on the Fokker-Planck equation prediction depends only on the residue of the PINN. In addition to the theoretical implications of this idea, the paper also shows numerical results on five rather well-known SDE systems that demonstrate the tightness of the built error bound.

**Strengths:**

- The paper touches a clearly novel problem and addresses it with an original solution. Particularly, the idea of building a recursion of excess losses is sensible and attractive. There exist similar approaches in other applications of learning theory. See for example a paper appearing at the upcoming NeurIPS:
https://arxiv.org/abs/2405.14681
- The formal analysis is rigorous and technically correct.
- The numerical analysis is well-conducted and well-communicated.

**Weaknesses:**

- Despite the title, I am having hard time locating how exactly the proposed approach helps bounding the propagation of error in SDE systems. I assume what is meant by propagation is propagation through time. The approach appears to develop bounds for standalone time points, ignoring how these errors warp when passed through the SDE kernel and accummulate.
- Another limitation of the approach is that it ignores the impact of the estimation error, which is a fundamental issue in quite many system identification problems especially under stochasticity. In the absence of a link to estimation error, it becomes a bit far-fetched to assess the real-world significance of the developed solution.
- The final weakness is the limited scope of the presented experiments. With due respect to the originality and rigor of the presented theoretical material as admitted above, I still think that the chosen set of SDEs is rather trivial. Specifically, their input dimensionality is too small and the level of difficulty is too limited to assess the significance of the presented solution. Three of the five SDE systems have analytical solutions, where the presented methods would not bring any value-added. I would start from dynamics such as the stochastic Lorenz attractor for which numerical solvers fail to deliver accurate results even in the deterministic variant. Only in such cases I can make sense of the true importance of developing rigorous recursive error bounds. I do expect that the presented material will also apply reasonably well to such cases but unfortunately those results are missing in the current version of the work.

**Questions:**

- I couldn’t locate the governing physics in Eq 4 mentioned in the description of Eq 6. The guidance from the authors is highly appreciated.
- Why do we need to set weights to $\mathcal{L}_0$ and $\mathcal{L}_r$? Is this not assigning arbitrarily relative importance to system initialization and system evolution? Should these weights not come from the theory itself if it is a complete theory? Is such weight association a common practice in the literature or a design choice made by this particular solution? How do the authors choose these weights in the experiments?

----
POST-REBUTTAL: The interaction with the reviewers made me conclude that the weaknesses I pointed out are indeed fundamental. I reflect this to my score by moving one step down from the initial borderline reject.

---

> ### Author Response · Authors · 2024-11-20
>
> **Q1**
> * The governing physics in Eq 4 is $\mathcal{D}[p] = 0$, where $\mathcal{D}[\cdot]$ is a compact notation of the Fokker-Planck PDE operator defined above Eq.4:
> $$\mathcal{D}[\cdot]:= \frac{\partial}{\partial t}[\cdot] + \sum_{i=1}^n \frac{\partial}{\partial x_i} [f_i \cdot] - \frac{1}{2} \sum_{i=1,j=1}^n \frac{\partial ^2}{\partial x_i \partial x_j} \left[ \sum_{k=1}^m g_{ik}g_{jk} \cdot \right].$$
>
> **Q2**
> * Yes, the weights determine the relative importance of the system's initial condition and evolution. These weights, typically chosen manually or via automatic tuning  (Karniadakis et al., 2021), affect training convergence but not the validity of our theorem. For all baseline experiments, we used constant weights $w_0=w_r=1$ (see Appendix B). Nevertheless, we emphasize that the key contribution of this paper is not the specifics of PINN training, which builds on existing methods, but the novel derivation of rigorous error bounds for the PINN.
>
> **Weakness 1**
> * We are uncertain about the exact interpretation of the title by the reviewer. To clarify, our work bounds the error $e(x,t) = p(x,t) - \hat{p}(x,t)$ between the Fokker-Planck solution $p(x,t)$ for a given SDE and the approximate PINN solution $\hat{p}(x,t)$ for all $t\in [0,T]$. The PINN directly approximates the PDF for $\mathbf{x}(t)$ without requiring the SDE kernel. We derive time-dependent bounds on $\max_x |e(x,t)|$ (Theorem 1, Corollary 2) and spatiotemporal bounds on $|e(x,t)|$ (Corollary 1). Hence, PINN $\hat{p}$ together with our error bounds explicitly account for the temporal evolution of $\mathbf{x}(t) \sim p(x,t) $ described by the SDE, addressing the primary theoretical challenge of this work. We are open to any suggestions to improve the title or clarify the paper further.
>
> **Weakness 2**
> * The reviewer is correct that this work assumes a given SDE; the process of obtaining the SDE or addressing potential modeling errors in its derivation is outside the scope of this study.
>
> **Weakness 3**
> * Thank you for this comment. We acknowledge the limited scope of the presented experiments and, following Reviewer 7jTR's suggestion, conducted additional experiments on 3D to 10D Ornstein-Uhlenbeck systems. The results are provided in the official comment above.  They demonstrate that our method scales well to 10D systems while maintaining tight error bounds.
>
> * We are also running experiments on a Lorenz63 system as suggested by the reviewer. As soon as the results become available, we'll add them to the paper.  But, we'd like to note that while training a PINN can take time, the primary bottleneck is in obtaining the "true" PDF for this system, which is needed for validating our error bounds. Due to the highly oscillatory nature of the system, accurate simulation requires an extremely small integration time step, making it computationally very expensive (e.g., a Monte Carlo simulation of duration 0.3 seconds takes over 36 hours using $10^9$ samples and 0.001 seconds integration time step). Hence, we anticipate it will take more than a week to obtain the first results. Nevertheless, we are keen on adding this case study to the paper as it'll show the power of the framework.
>
> * Finally, we wish to emphasize the significance of the results on lower-dimensional systems. A major challenge in such cases is obtaining an accurate "true" PDF solution via numerical integration. Although the 1D/2D nonlinear SDEs are less complex than the suggested 3D Lorenz dynamics, their state distributions and uncertainties over time lack a closed-form solution. To validate the error bounds, we simulated these nonlinear systems with $10^9$/$10^8$ samples and 0.0005/0.01 seconds integration time step, requiring 100 hours and 13 hours, respectively, to compute the "true" PDF for the 1D nonlinear SDE and the 2D inverted pendulum experiments.  In contrast, as shown in Table 2, our method required only 1.3 hours and 1.4 hours, respectively, to predict the PDF evolution and construct the error bounds. For completeness, we will add the computation times for obtaining the "true" PDF to Table 1 and provide further details in Appendix B.

---

> > ### Comment · Reviewer_tHQ1 · 2024-11-22
> > **No change**
> >
> > Thanks for the additional experiments. While being higher-dimensional, Ornstein-Uhlenbeck (also its time-varying version) is still not a sort of dynamical system where a learning-based approach will bring a meaningful benefit. These are setups where classical non-machine-learning-based SDE methods work straightforwardly. In such use cases there is only approximation error, but no estimation error, hence they are sterile to the biggest headaches of machine learning research. I still keep my opinion that the problem scope is chosen in a way to significantly limit the potential value of the contribution. I update my borderline score to a more decisive one.

---

> ### Author Response · Authors · 2024-11-23
> **Importance of the Problem and additional chaotic SDE experiment**
>
> We thank the reviewer for the prompt reply. We would like to highlight two points: **importance of the considered problem** and **demonstration of the capability of the proposed method through additional experiment on a chaotic SDE**.
>
>
> **Importance of the problem**
>
> We would like to stress that even though we assume a known SDE, the problem considered in this paper remains highly significant. In particular, propagating the uncertainty of SDEs is a major bottleneck in various fields, including control theory (Ridderhof et al.,
> 2019; Oguri \& McMahon, 2021), study of dynamical systems  (Sun \& Kumar, 2016; Khatri \&
> Scheeres, 2023; Figueiredo et al., 2024), and analysis of PINNs  (De Ryck \& Mishra, 2022b;a; Mishra \& Molinaro, 2023; De Ryck et al., 2024).
>
> Furthermore, we respectfully disagree with the statement that, in our experimental results, the use of "a learning-based approach does not bring a meaningful benefit." In fact, for all the nonlinear systems considered, our approach offers two significant advantages:
> 1. **Speedup:** Our method achieves a speedup of 6–75 times compared to existing Monte Carlo-based approaches.
> 2. **Error bound:** Traditional approaches, including Monte Carlo, moment-based, and Gaussian-mixture-based methods, do not provide explicit error bounds for the full probability distribution. In contrast, our deep learning-based approach delivers worst-case error bounds that are both tight and practically valuable.
>
> **Additional Experiment**
>
> We present additional experiment to demonstrate the effectiveness of our approach on challenging dynamics: the stochastic Duffing oscillator, which exhibits chaotic behavior. We kindly refer the reviewer to the recent work (Anderson \& Farazmand, 2024) for a discussion on the difficulties of dealing with such systems. Notably, the approach presented in that paper cannot handle general SDEs and, more importantly, does not provide error bounds.
>
> To obtain the "true" PDF for the stochastic Duffing oscillator, we used a time step of $0.005$ seconds and $10^7$ samples for the Monte Carlo simulation, which required **6.3 hours**. In comparison, our approach took only **1.1 hours** to construct the approximate solution while also quantifying error bounds.
>
> | System | Dynamics | I.C. | True Solution |
> |:--|:--|:--|:--|
> | Stochastic Duffing Oscillator    | $dx= \begin{bmatrix} x_2 \\ 1.0 x_1 -0.2 x_2 -1.0 x_1^3 \end{bmatrix}dt + \begin{bmatrix} 0.0 & 0.0 \\ 0.0 & 1/\sqrt{20} \end{bmatrix}dw$ | $\mathcal{N}(\begin{bmatrix} 1.0 \\ 1.0 \end{bmatrix}, \begin{bmatrix} 0.05 & 0.0 \\ 0.0 & 0.05 \end{bmatrix})$ | Monte-Carlo (6.3 hrs)            |
>
> | System | $\hat{p}$ loss | $\hat{e}_1$ loss | $t_{\text{train}}^{\hat{p}}$ (sec) | $t_{\text{train}}^{\hat{e}_1}$ (sec) | $e_S^{\max}$ | $e_S^{\text{avg}}$ | $\text{Gap}^{\min}$ | $\text{Gap}^{\max}$ | $\alpha_1^{\max}$ | $\alpha_1^{\text{var}}$ |
> |:--|:--:|:--:|:--:|:--:|:--:|:--:|:--:|:--:|:--:|:--:|
> | Stochastic Duffing Oscillator                            | $4 \times 10^{-3}$ | $2 \times 10^{-4}$ | 358                         | 3906                          | 0.24         | 0.20               | 0.079               | 0.123            | 0.444             | $1 \times 10^{-2}$     |

---

> > ### Comment · Reviewer_tHQ1 · 2024-11-28
> >
> > Thanks for your effort. We completely agree on the importance of quantification of the SDE uncertainties propagated through time. We don't agree on the meaning of the information conveyed by an experiment conducted on SDE systems where the uncertainty propagation characteristics are perfectly known, i.e. not only the drift and the diffusion dynamics have an analytical solution. Duffing oscillator is a good step forward, thanks for it. But now this is the only experiment I have to make decision, which I don't think is sufficient. It is after all a one-dimensional system. Dropping down to 1.1 hours from 6.3 is nice but let's remember that the comparison is with respect to the naive MC simulation approach. How about established methods such as
> >
> > Archambeau et al., Gaussian Process Approximations of Stochastic Differential Equations, 2007
> >
> > Or any other approach one can name as a true representative of the state of the art in approximating challenging nonlinear SDE systems?

---

> ### Author Response · Authors · 2024-11-30
> **Clarifying Misinterpretations and Further Explanation of Key Aspects of Our Work**
>
> We thank the reviewer once again for their response and for staying engaged. We would like to clarify certain misinterpretations regarding our experiments and emphasize that the primary novelty of our work lies in the derivation of the error bounds.
>
> # Regarding Experiments
>
> We would like to emphasize the following key points regarding our experiments:
>
>
> 1. **Closed-Form Solutions:**
>
>    - Closed-form (analytical) solutions do not exist for the 1D/2D nonlinear SDEs considered in our experiments.
>
>    - For propagating PDFs of time-varying OU SDEs, numerical integration is required, which is prone to error accumulation
>
>    - In contrast, our approach is faster and provides explicit error bounds. In fact, if we only compare the computation time to generate PDF between accurate Monte-Carlo and our approach, we reduced the time from {100, 13, 6.3 hours} to {0.2, 0.4, 0.1 hours} for the {1D, 2D inverted pendulum, Duffing oscillator} experiments, achieving speedups of up to 500 times.
>
>
> 2. **Duffing Oscillator Dimensionality:**
>
>    - The Duffing oscillator SDE is a **2-dimensional** system, not 1D as mentioned by the reviewer.
>
> # Related work
>
> 1. **Regarding the Work by Archambeau et al. (2007):**
>
>    - The method proposed in that paper:
>      - Requires **observations** of the process (i.e., data from the actual system)
>      - Provides only an approximation to the posterior measure.
>      - Relies on Gaussian Processes, which are known to struggle with high-dimensional problems.
>
>
>    - In contrast, our approach:
>      - Does not require data (observations) from the system. Instead, it is physics-informed and based on the Fokker-Planck PDE.
>      - Provides explicit error bounds.
>      - Scales effectively to high-dimensional problems, as demonstrated on 10D OU systems.
>
>
> 2. **Other Recent Work:**
>
>    - Work by **Yashica Khatri and Daniel J. Scheeres (2023)** uses the Gaussian Mixture Model (GMM) method to propagate uncertainties in SDEs. However, that work is also an approximate approach and lacks explicit error bounds.
>    - In contrast, our method provides explicit error bounds.
>    - **Note:** Our method of bounding error can be used to quantify the error in the GMM method by *Khatri and Scheeres (2023)*, as our approach is general and only requires an approximation of the PDF.
>
>
> # Key Contributions & Novelty
> We further emphasize that our work makes two key contributions:
>
>    1. **Application of Deep Learning for PDF Propagation of SDEs:** Our approach is physics-informed and does not require data.
>
>    2. **Derivation of Error Bounds:** The majority of the novelty lies in this aspect, involving theoretical derivations that pave the way for using neural networks in safety-critical domains.
>
>
> Finally, we note that improving the training process is an area of ongoing work.

---

### Author Response · Authors · 2024-11-20
**Additional experiments: high-dimensional SDEs**

A common concern raised by the reviewers is applicability of our method to high-dimensional systems. To address this, we conducted high-dimensional (time-varying) Ornstein-Uhlenbeck experiments, suggested by Reviewer 7jTR. The results are summarized below, which will be added to the paper. Additional details of the OU experiments are provided in Appendix B.6.

In summary, we observe that:
* **Scalability:** our framework is able to scale to 10D time-varying OU.
* **Stability:** the variance on $\alpha_1$ over the time domain is small, showing the error bound's applicability for all time.
* **Training Challenges:** training of $\hat{e}_1$ may encounter local minima due to random initialization of neural networks.
* **$\alpha_1$ Condition:** $\alpha_1 < 1$ is satisfied in all experiments, though it becomes increasingly challenging to meet as dimensionality grows.
* **Error bound tightness:** the error bounds are fairly tight across all cases.

**Additional Update of Table 1**
(see the revised paper for the complete Table 1)
| System | Dynamics | I.C. | True Solution |
|:--|:--|:--|:--|
| 1D Nonlinear SDE         | $dx = (-0.1x^3+0.1x^2+0.5x+0.5)dt + 0.8 dw$                                      | $\mathcal{N}(-2, 0.5^2)$                               | Monte-Carlo (100 hrs)           |
| Inverted Pendulum SDE    | $dx= \begin{bmatrix} x_2 \\ -\sin(x_1) \end{bmatrix}dt + \begin{bmatrix} 0.5 & 0.0 \\ 0.0 & 0.5 \end{bmatrix}dw$ | $\mathcal{N}(\begin{bmatrix} 0.5 \pi \\ 0.0 \end{bmatrix}, \begin{bmatrix} 0.5 & 0.0 \\ 0.0 & 0.5 \end{bmatrix})$ | Monte-Carlo (13 hrs)            |
| 3D OU                    | $dx = (A_3 x) dt + B_3 dw$                                                        | $\mathcal{N}(\mu_3, \Sigma_3)$                         | Numerical integration           |
| 3D Time-varying OU       | $dx = (\tilde{A}_3(t) x) dt + B_3 dw$                                                    | $\mathcal{N}(\mu_3, \Sigma_3)$                       | Numerical integration           |
| 7D OU                     | $dx = (A_{7} x) dt + B_{7} dw$                                                   | $\mathcal{N}(\mu_{7}, \Sigma_{7})$                     | Numerical integration           |
| 10D OU                    | $dx = (A_{10} x) dt + B_{10} dw$                                                 | $\mathcal{N}(\mu_{10}, \Sigma_{10})$                   | Numerical integration           |
| 10D Time-varying OU      | $dx = (\tilde{A_{10}}(t) x) dt + B_{10} dw$                                             | $\mathcal{N}(\tilde{\mu_{10}}, \Sigma_{10})$                 | Numerical integration           |


**Additional Update of Table 2**
(see the revised paper for the complete Table 2. Note that the computation time for $t_{\text{train}}^{\hat{p}}$ and $t_{\text{train}}^{\hat{e}_1}$ is in seconds.)

| System | $\hat{p}$ loss | $\hat{e}_1$ loss | $t_{\text{train}}^{\hat{p}}$ | $t_{\text{train}}^{\hat{e}_1}$ | $e_S^{\max}$ | $e_S^{\text{avg}}$ | $\text{Gap}^{\min}$ | $\text{Gap}^{\max}$ | $\alpha_1^{\max}$ | $\alpha_1^{\text{var}}$ |
|:--|:--:|:--:|:--:|:--:|:--:|:--:|:--:|:--:|:--:|:--:|
| 3D OU                            | $1 \times 10^{-4}$ | $8 \times 10^{-3}$ | 276                         | 2017                          | 0.05         | 0.04               | 0.015               | 0.029               | 0.20               | $2 \times 10^{-4}$     |
| 3D Time-varying OU               | $1 \times 10^{-4}$ | $4 \times 10^{-3}$ | 338                         | 2219                          | 0.06         | 0.05               | 0.020               | 0.032               | 0.16               | $3 \times 10^{-4}$     |
| 7D OU                            | $2 \times 10^{-4}$ | $1 \times 10^{-2}$ | 1018                        | 2684                          | 0.19         | 0.11               | 0.036               | 0.098               | 0.74               | $2 \times 10^{-2}$     |
| 10D OU                           | $1 \times 10^{-4}$ | $1 \times 10^{-2}$ | 1710                        | 3670                          | 0.20         | 0.15               | 0.067               | 0.119               | 0.68               | $5 \times 10^{-3}$     |
| 10D Time-varying OU              | $1 \times 10^{-4}$ | $6 \times 10^{-3}$ | 2835                        | 13883                         | 0.16         | 0.12               | 0.053               | 0.095               | 0.98               | $9 \times 10^{-3}$     |

*Footnote:* For the 1D Nonlinear SDE on GPU, the variance of $\alpha_1$ over all random seeds $i=\{0,1,...,5\}$ is $\text{var}_{t,i} \alpha^{(i)}_1(t) = 0.11$.

---

> ### Author Response · Authors · 2024-12-01
> **Additional experiment: 2D Stochastic Duffing Oscillator**
>
> We present additional experiment to demonstrate the effectiveness of our approach on challenging dynamics: the 2-dimensional stochastic Duffing oscillator, which exhibits chaotic behavior. We kindly refer the reviewers to the recent work (Anderson \& Farazmand, 2024) for a discussion on the difficulties of dealing with such systems. Notably, the approach presented in that paper cannot handle general SDEs and, more importantly, does not provide error bounds.
>
> To obtain the "true" PDF for the stochastic Duffing oscillator, we used a time step of $0.005$ seconds and $10^7$ samples for the Monte Carlo simulation, which required **6.3 hours**. In comparison, our approach took only **0.1 hours** to construct the approximate PDF solution and **1 hour** for the error bounds.  We will also add these results to the paper.
>
> | System | Dynamics | I.C. | True Solution |
> |:--|:--|:--|:--|
> | Stochastic Duffing Oscillator    | $dx= \begin{bmatrix} x_2 \\ 1.0 x_1 -0.2 x_2 -1.0 x_1^3 \end{bmatrix}dt + \begin{bmatrix} 0.0 & 0.0 \\ 0.0 & 1/\sqrt{20} \end{bmatrix}dw$ | $\mathcal{N}(\begin{bmatrix} 1.0 \\ 1.0 \end{bmatrix}, \begin{bmatrix} 0.05 & 0.0 \\ 0.0 & 0.05 \end{bmatrix})$ | Monte-Carlo (6.3 hrs)            |
>
> | System | $\hat{p}$ loss | $\hat{e}_1$ loss | $t_{\text{train}}^{\hat{p}}$ (sec) | $t_{\text{train}}^{\hat{e}_1}$ (sec) | $e_S^{\max}$ | $e_S^{\text{avg}}$ | $\text{Gap}^{\min}$ | $\text{Gap}^{\max}$ | $\alpha_1^{\max}$ | $\alpha_1^{\text{var}}$ |
> |:--|:--:|:--:|:--:|:--:|:--:|:--:|:--:|:--:|:--:|:--:|
> | Stochastic Duffing Oscillator                            | $4 \times 10^{-3}$ | $2 \times 10^{-4}$ | 358                         | 3906                          | 0.24         | 0.20               | 0.079               | 0.123            | 0.444             | $1 \times 10^{-2}$     |

---

### Meta-Review · Area_Chair_ZxKk · 2024-12-19

**Metareview:**

The paper 'Error Bounds for Deep Learning-based Uncertainty Propagation in SDEs' was reviewed by 4 reviewers who gave it an average score of 5.0 (final scores: 3+5+6+6). The reviewers appreciated the practical nature of the problem addressed by the authors. However, several reviewers express concerns related to the rather simplistic view on the error estimation problem and raise concerns related to the practicality of the approach. Even if some of the reviewers consider this paper a borderline accept, the overall view is that the paper does not quite meet the bar for acceptance in its current form.

**Additional Comments On Reviewer Discussion:**

The authors provided rebuttals, and the majority of the reviewers were active during the discussion phase. However, the average score decreased from 5.25 -> 5.0 during the discussion, because one reviewer decreased their score to a more decisive one while another increased it moderately.

---

### Decision · Program_Chairs · 2025-01-22

Reject